# Redirection of SARS-CoV-2 to phagocytes by intranasal sACE2-Fc as a universal decoy confers complete prophylactic protection

Jingyi Wang[1†], Jiangchuan Li[1†], Alex WH Chin[2,3†], Bin Luo[1,4], Junkang Wei[1], Jiale Qiu[1], Jianwei Ren[1,4], Yin Xia[1], Thomas Braun[1,5], Leo LM Poon[2,3,6]*, Bo Feng[1,4,7]*

[1]School of Biomedical Sciences, Faculty of Medicine; GIBH CAS-CUHK Joint Research Laboratory on Stem Cell and Regenerative Medicine, The Chinese University of Hong Kong, Hong Kong, China; [2]Centre for Immunology and Infection, Hong Kong Science and Technology Park, Hong Kong, China; [3]School of Public Health, LKS Faculty of Medicine, The University of Hong Kong, Hong Kong, China; [4]Centre for Regenerative Medicine and Health, Hong Kong Institute of Science and Innovation, Chinese Academy of Sciences, Hong Kong, China; [5]Department of Cardiac Development and Remodeling, Max-Planck-Institute for Heart and Lung Research, Bad Nauheim, Germany; [6]HKU-Pasteur Research Pole, LKS Faculty of Medicine, The University of Hong Kong, Hong Kong, China; [7]Guangzhou Institutes of Biomedicine and Health, Chinese Academy of Sciences, Guangzhou, China

*For correspondence:
llmpoon@hku.hk (LLMP);
fengbo@cuhk.edu.hk (BF)

[†]These authors contributed equally to this work

Competing interest: The authors declare that no competing interests exist.

## eLife Assessment

This manuscript presents a **valuable** antiviral approach using an engineered ACE2-Fc fusion protein that demonstrates broad-spectrum neutralization capacity against SARS-CoV-2 variants and achieves significant prophylactic protection in animal models through a novel Fc-mediated phagocytosis mechanism. The study provides **convincing** evidence for protective efficacy through rigorous in vivo validation in mice, mechanistic characterization via transcriptomic analysis and biodistribution studies, and demonstration of antibody-dependent cellular phagocytosis as the primary clearance mechanism mediated by the decoy. The work will be of interest to researchers working in vaccine development and associated immune responses.

**Abstract** The rapid evolution of SARS-CoV-2 and other respiratory RNA viruses limits the success of current vaccines and antibody-based therapies. Engineered decoy receptors based on soluble angiotensin-converting enzyme 2 (sACE2) offer promising alternatives but show limited clinical success. This study conducted functional and mechanistic analyses using an optimized sACE2 mutant fused to human IgG1 Fc (B5-D3) as a representative, revealing redirection of virus–decoy complexes from epithelial infection to lysosomal degradation in phagocytes beyond viral neutralization. Intranasal prophylactic delivery of B5-D3 confers complete protection in SARS-CoV-2-infected K18-hACE2 mice, regardless of age. Abrogation of Fc effector functions compromises antiviral protection, indicating that Fc-mediated uptake of virus–decoy complexes is critical. Transcriptomic analysis suggests that B5-D3 induces early immune activation in the lungs of infected mice. Biodistribution and flow cytometry reveal selective targeting of airway phagocytes. In vitro assays confirm lysosomal degradation of virus–decoy complexes by macrophages without productive

infection. These findings reveal a distinct antiviral mechanism via phagocytic clearance, supporting refined regimens for decoy treatments against SARS-CoV-2 and potentially other respiratory viruses.

## Introduction

The incessant evolution of severe acute respiratory syndrome coronavirus 2 (SARS-CoV-2) and frequent breakthrough infections during the coronavirus disease 2019 (COVID-19) pandemic underscore the critical need for effective antiviral strategies that are less susceptible to immune escape than conventional vaccines and monoclonal antibody (mAb) therapies (*Wang et al., 2023a*).

Soluble angiotensin-converting enzyme 2 (sACE2) therapies, which employ recombinant forms of the human angiotensin-converting enzyme 2 (ACE2) receptor—the primary binding site for the SARS-CoV-2 spike protein (*Wang et al., 2023a*; *Hoffmann et al., 2020*; *Ozono et al., 2021*; *Ramanathan et al., 2021*; *Li et al., 2022*)—as viral decoys, have emerged as a promising alternative (*Monteil et al., 2020*). However, an early clinical version (amino acid [aa] 1–740, APN01) showed limited therapeutic benefit (*Zoufaly et al., 2020*) and raised safety concerns about interference with the endogenous renin–angiotensin system (RAS) (*Santos et al., 2018*). Subsequent protein engineering greatly improved the pharmacological properties of sACE2, including fusion with a human IgG1 Fc domain (sACE2-Fc) to enhance serum half-life (*Liu et al., 2018*), and mutagenesis to enhance spike-binding affinity (*Payandeh et al., 2020*; *Lei et al., 2020*; *Glasgow et al., 2020*) and abolish enzymatic activity (*Payandeh et al., 2020*; *Glasgow et al., 2020*; *Liu et al., 2020*). Potent sACE2-Fc mutants have shown broad-spectrum neutralization against SARS-CoV-2 variants in animal models (*Chen et al., 2022*; *Hassler et al., 2023*; *Kober et al., 2024*). However, their efficacy in protecting hosts from viral infections was often incomplete. Despite the evidence suggesting a role for Fc-mediated effector functions in sACE2-Fc efficacies (*Chen et al., 2022*), underlying immune mechanisms remained poorly understood. Further investigation that systematically assesses the potential to optimize decoy design, strategies of administration, and mechanisms of action is pivotal to the development of ACE2 decoy-based antivirals and harnessing their full potential.

In this study, we engineered a potent yet minimally mutated sACE2-Fc decoy candidate (B5-D3) with just two mutations to enhance spike-binding (T92Q) while eliminating enzymatic activity (H374N). Broad-spectrum neutralization capacity against multiple SARS-CoV-2 variants was confirmed by in vitro neutralization assays. Markedly, stepwise examinations of various administration routes and time points identified intranasal (IN) prophylaxis as the most effective regimen for B5-D3, which conferred complete protection against SARS-CoV-2 infection in K18-hACE2 mice across age groups. Whereas B5-D3 intravenously administered either prior to or post infection showed activity that moderately improved disease outcome. To understand how sACE2-Fc decoys influence viral fate and achieve superior antiviral protection via IN prophylaxis, we carried out systematic, mechanistic investigations through transcriptomics, bio-distribution, and phagocytosis analysis. Our results revealed that IN-delivered B5-D3 engages airway phagocytes to promote early viral clearance and host immune activation, which uncovers a distinct antiviral mechanism and offers a universal and commonly applicable 'decoy strategy' to combat unknown air-borne respiratory viruses in the future.

## Results

### Engineered sACE2-Fc decoys with two single mutations achieve robust neutralization against SARS-CoV-2 variants

To generate representative ACE2 decoys with optimal performance, we adopted the established sACE2-Fc fusion design (*Liu et al., 2018*; *Figure 1A*, *Figure 1—figure supplement 1*) and selectively verified mutation(s) proposed in prior studies, either to enhance the binding of human ACE2 to SARS-CoV-2 spikes (*Chan et al., 2020*) (B2–B6) or to abolish enzymatic activity (*Payandeh et al., 2020*; *Lei et al., 2020*; *Glasgow et al., 2020*; *Guy et al., 2005*) (A2, A3, D1–D5) (*Figure 1—figure supplement 2A, B*). The sACE2-Fc candidates generated with either type of mutation(s) were all verified for pseudovirus-based neutralization assays (*Crawford et al., 2020*) and ACE2 enzymatic activity. Indeed, mutants B2–B6 showed consistently enhanced neutralization capacity against both Wuhan-Hu-1 and D614G pseudoviruses (*Korber et al., 2020*; *Figure 1B*, *Figure 1—figure supplement 2C*,

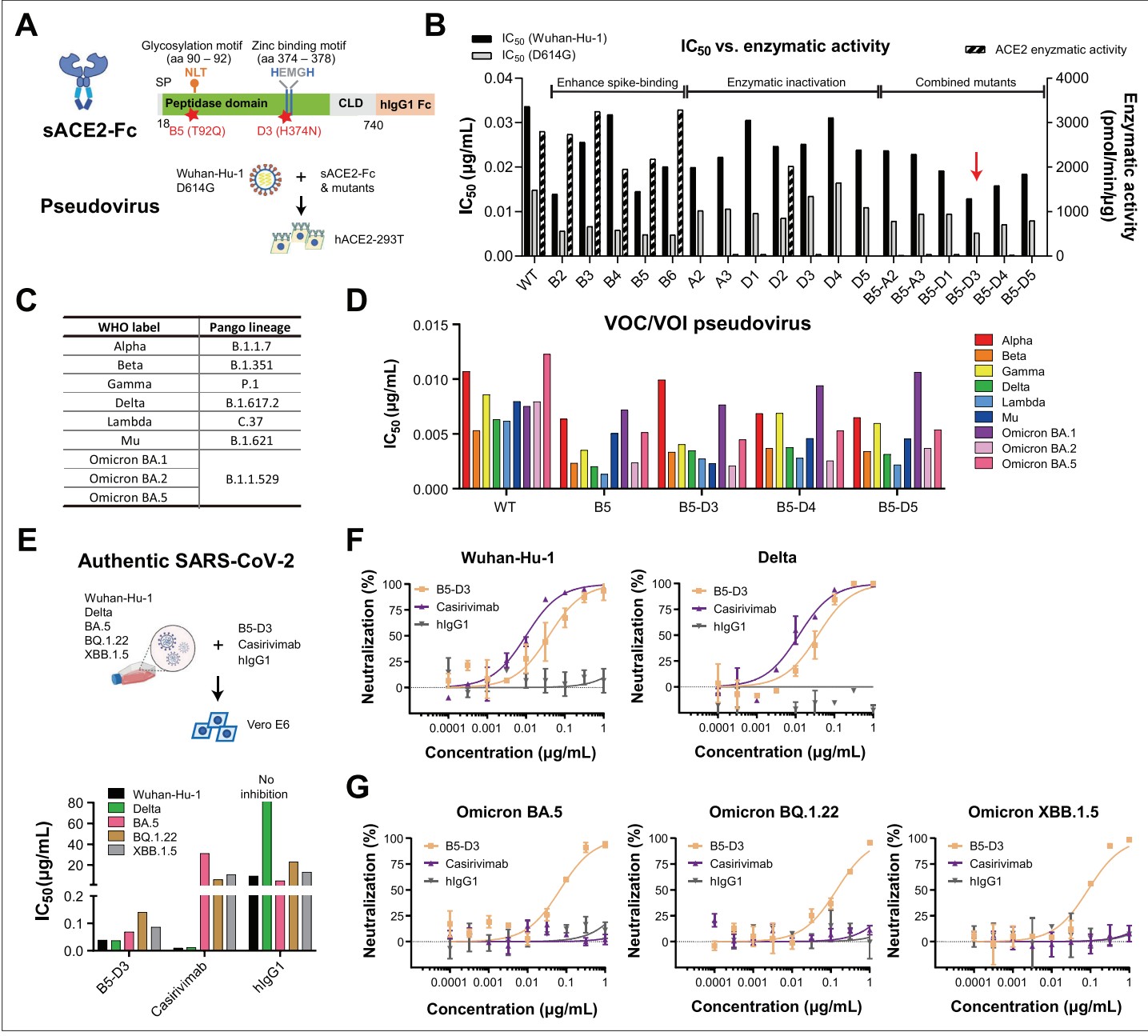

**Figure 1.** Enhanced sACE2-Fc with two single mutations exhibited broad-spectrum neutralization of SARS-CoV-2 variants. (**A**) Schematic representation of sACE2-Fc structure (upper) and neutralization assay setup (lower). Key amino acid positions (90–92 and 374–378) involved in glycosylation and zinc binding are highlighted. Red stars mark the positions of mutations in the sACE2-Fc mutant B5-D3. SP, signal peptide; CLD, collectin-like domain; hIgG1, human IgG1. (**B**) Comparative bar graph showing the half-maximal inhibitory concentration (IC$_{50}$) values for neutralization of Wuhan-Hu-1 and D614G pseudoviruses by WT sACE2-Fc and mutants (B2 to B6, A2, A3, D1 to D5, and B5-derivatives). The red arrow emphasizes the superior performance of the B5-D3 mutant. Enzymatic activity of each construct is plotted on the right axis. (**C**) List of pseudoviruses carrying spikes from different SARS-CoV-2 variants tested, categorized by the World Health Organization (WHO) into VOCs and VOIs. (**D**) Graph displaying IC$_{50}$ values of WT sACE2-Fc, B5, and B5-D3/4/5 mutants against various SARS-CoV-2 VOCs and VOIs in neutralization assays. (**E**) Schematics of the plaque-reduction neutralization tests (PRNTs) process (upper) and the resulting IC$_{50}$ values for B5-D3, Casirivimab, and hIgG1 against authentic SARS-CoV-2 (lower). (**F,G**) Dose–response curves depicting the neutralization efficacy of B5-D3 (orange), Casirivimab (purple), and hIgG1 (gray) in PRNTs against authentic SARS-CoV-2 Wuhan-Hu-1 and Delta strains (**F**), and Omicron sub-lineages (**G**). Data are presented as mean ± standard deviation (SD) from duplicate experiments.

The online version of this article includes the following figure supplement(s) for figure 1:

**Figure supplement 1.** Establishment of the pseudoviral infection platform and generation of sACE2 decoys.

*Figure 1 continued on next page*

**Figure supplement 2.** Engineering and characterization of enhanced sACE2 decoys.

**Figure supplement 3.** Broad-spectrum neutralization against pseudoviruses of SARS-CoV-2 VOC/VOI strains by sACE2-Fc candidate mutants.

**Figure supplement 4.** AAV-delivered prolonged overexpression of WT sACE2-Fc and candidate mutants in K18-hACE2 mice.

*D*). Notably, B5 with the single T92Q mutation, which increases spike affinity by removing a critical glycosylation site at N90 (*Suryamohan et al., 2021*), exhibited remarkable neutralization enhancement among other multi-mutants. Meanwhile, mutations selected for catalytic inactivation, including the previously reported mutation pairs A2 (*Lei et al., 2020*; *Guy et al., 2005*), A3 (*Payandeh et al., 2020*), single mutations derived from these pairs (D1, D3, D4, but not D2), and an individually reported single mutation D5 (*Glasgow et al., 2020*), effectively abolished enzymatic activity while showing minimal effect on spike binding (*Figure 1B*, *Figure 1—figure supplement 2E*). We next combined the B5 (T92Q) mutation with each of the inactivating mutations. Notably, among the resulting compound mutants, B5-D1, B5-D3, B5-D4, and B5-D5 with two mutations remained enzymatically inactive while retaining comparable or stronger neutralization capacity than B5-A2 and B5-A3 with three mutations (*Figure 1B*, *Figure 1—figure supplement 2F–H*). B5-D3 (T92Q/H374N) emerged as one of the best candidates (*Figure 1A, B*), exhibiting minimal deviation from wild-type (WT) ACE2 in structural modeling (root mean square deviation [RMSD] = 0.212 Å; *Figure 1—figure supplement 2I*; *Abramson et al., 2024*).

To assess the breadth of neutralization, we tested three double mutants (B5-D3, B5-D4, and B5-D5) against pseudoviruses bearing spikes from various variants of concern (VOCs) and variants of interest (VOIs) (*Wang et al., 2023a*; *Garcia-Beltran et al., 2021*; *Mlcochova et al., 2021*; *Kimura et al., 2022*; *Halfmann et al., 2022*; *Cao et al., 2022*). All three designs showed dose-dependent neutralization with higher potency than WT sACE2-Fc (*Figure 1C, D*; *Figure 1—figure supplement 3*). We further examined B5-D3, as a representative decoy candidate, against authentic SARS-CoV-2 using plaque-reduction neutralization tests (PRNTs) in Vero E6 cells, which indeed confirmed its robust activity against Wuhan-Hu-1, Delta, and Omicron variants BA.5, BQ.1.22, and XBB.1.5 strains (*Wang et al., 2023a*; *Mlcochova et al., 2021*; *Wang et al., 2023b*; *Amin et al., 2023*; *Figure 1E–G*). In contrast, Casirivimab, serving as a positive control (*Weinreich et al., 2021*), showed efficacy only against early variants (Wuhan-Hu-1 and Delta; *Figure 1F*), but failed to neutralize Omicron sublineages (*Figure 1G*). These results demonstrate that a rationally engineered sACE2-Fc decoy with only two mutations could achieve potent and safe neutralization across SARS-CoV-2 variants, reducing the potential risks associated with extensive mutagenesis.

## Prolonged in vivo overexpression of sACE2-Fc double mutants demonstrates minimal RAS disturbance and no tissue damage in mice

Next, we evaluated the safety of the sACE2-Fc double mutants in vivo (*Figure 1—figure supplement 4A*). Adult K18-hACE2 transgenic mice with immune tolerance to human ACE2 (*McCray et al., 2007*) were intravenously injected with adenovirus-associated virus (AAV) vectors encoding either WT sACE2-Fc or double mutants at a dose of $1 \times 10^{11}$ genome copies (GC) per mouse. Notably, serum levels of the double mutants (B5-D3, B5-D4, and B5-D5) were significantly higher than those of WT sACE2-Fc (*Figure 1—figure supplement 4B* and *Supplementary file 1a*). This trend was further supported by quantification of AAV genomes in the liver, indicating greater in vivo stability or tolerance of the double mutants (*Figure 1—figure supplement 4C*).

Importantly, despite prolonged high-level expression, ELISA measurements of serum renin, Angiotensin II (Ang II), and Ang (1-7) (*Wang et al., 2023a*; *Hoffmann et al., 2020*; *Ozono et al., 2021*; *Ramanathan et al., 2021*; *Li et al., 2022*; *Monteil et al., 2020*; *Zoufaly et al., 2020*; *Santos et al., 2018*) demonstrated minimal disturbance to the RAS in mice treated with the double mutants (*Figure 1—figure supplement 4D–F*). In contrast, WT sACE2-Fc treatment led to significantly elevated serum levels of renin and Ang II, indicating a disruption of the RAS (*Figure 1—figure supplement 4D, E*). Histological examination of multiple organs at the end point showed no evidence of tissue damage in any of these groups (*Figure 1—figure supplement 4G*). These observations collectively underscore the improved safety of catalytically inactive sACE2-Fc mutants, supporting their suitability for prolonged or repeated use.

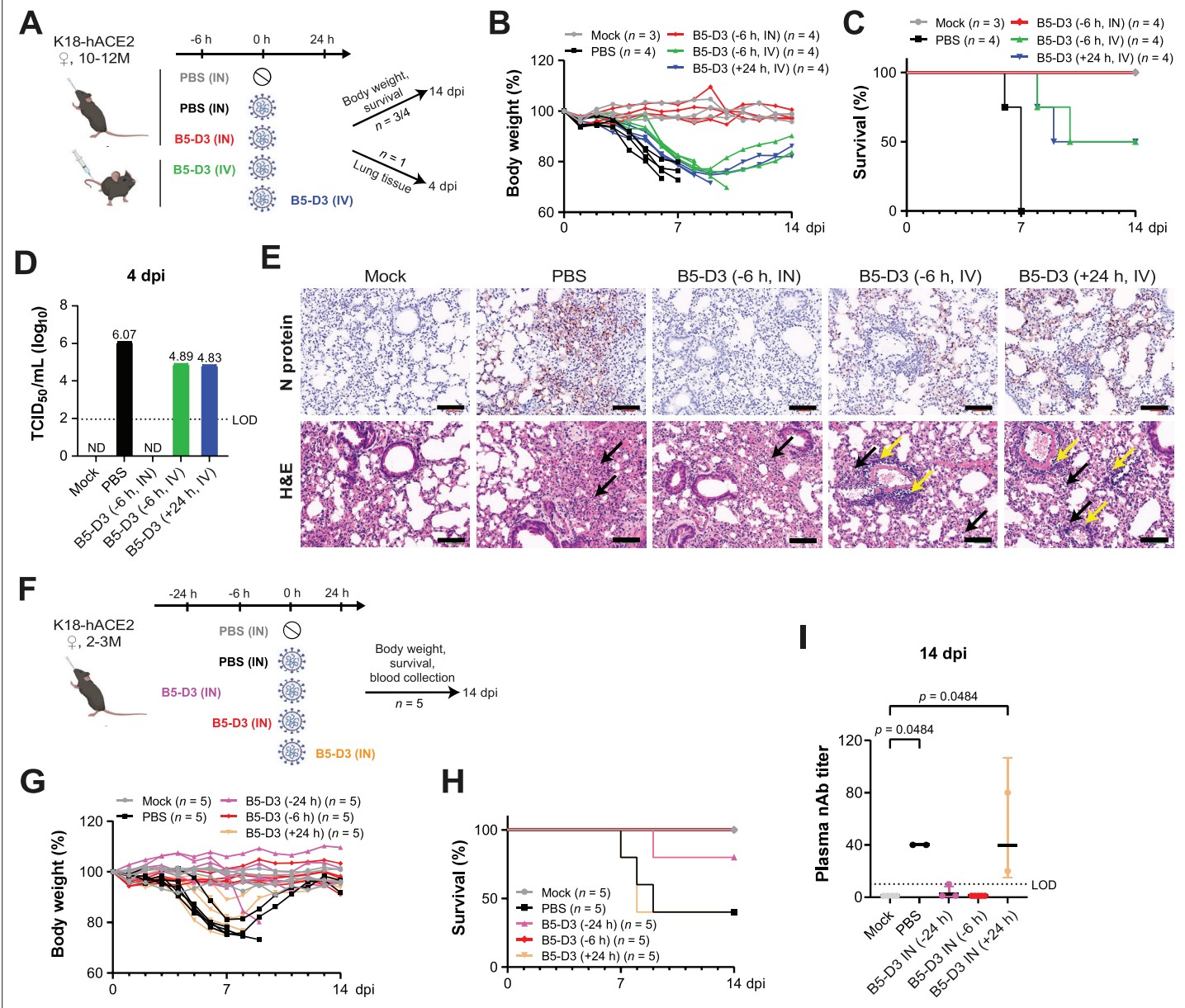

**Figure 2.** Enhanced survival and reduced infection in K18-hACE2 mice through intranasal prophylaxis with B5-D3 against SARS-CoV-2. (**A–E**) Female K18-hACE2 mice, aged 10–12 months, were inoculated with $1 \times 10^4$ PFU of SARS-CoV-2 (Wuhan-Hu-1 strain). Mice were treated with B5-D3 6 hr prior (–6 hr) via intranasal (IN, red) or intravenous (IV, green) routes, or 24 hr post-infection (+24 hr, blue) via IV ($n = 4 + 1$). IN PBS administered 6 hr prior to viral challenge served as the vehicle control (black; $n = 4 + 1$), and PBS alone was used for mock control (gray; $n = 3 + 1$) (**A**). Body weight and survival ($n = 3$ or 4) were monitored over 14 days (**B, C**). One mouse from each group was sacrificed at 4 dpi for analysis of viral titers in lung homogenates using a median tissue culture infectious dose (TCID$_{50}$) assay (**D**) and histological analysis of lung sections (upper, IHC staining for N protein; lower, H&E staining) (**E**). Black arrows indicate alveolar thickening, and yellow arrows show leukocyte infiltration. Scale bar = 100 μm. ND, not detected; LOD, limit of detection. (**F–I**) Young female K18-hACE2 mice, aged 2–3 months, were inoculated similarly and treated with B5-D3 via IN route at 24 hr before (–24 hr, pink), 6 hr before (–6 hr, red), or 24 hr after (+24 hr, orange) the viral challenge ($n = 5$). Mice receiving IN PBS 6 hr before infection served as the vehicle control (black), with mock control mice receiving PBS alone (gray) (**F**). Body weight (**G**) and survival (**H**) were recorded for 14 days. Neutralizing antibody titers against Wuhan-Hu-1 in serum samples from surviving mice at 14 dpi were determined using Vero E6 cells (**I**). h, hour(s). dpi, day(s) post injection. nAb, neutralizing antibody. Data are presented as the geometric mean ± geometric SD. Statistical significance was determined using Dunn's multiple comparisons test.

## Prophylactic administration of the sACE2-Fc B5-D3 mutant via the intranasal route exhibits superior protection against SARS-CoV-2

Next, we evaluated the in vivo efficacy of the sACE2-Fc double mutant B5-D3 against SARS-CoV-2 infection using aged K18-hACE2 mice (10–12 months old) (*Figure 2A*). Six hours before inoculating with $1 \times 10^4$ plaque-forming unit (PFU) of SARS-CoV-2 (Wuhan-Hu-1 strain), mice received a prophylactic dose of recombinant B5-D3 protein either intranasally (IN, 2.5 mg/kg) or intravenously (IV, 15 mg/kg). To simulate a therapeutic intervention, an additional group received IV B5-D3 (15 mg/kg) 24 hr post-virus inoculation. The vehicle control group received an intranasal PBS administration 6 hr before viral challenge. Over a 14-day observation period, all mice in the PBS group exhibited significant weight loss and succumbed to infection by 7 days post-infection (dpi) (*Figure 2B, C*, black lines). Both IV-treated groups exhibited initial weight loss similar to the PBS group; however, two out of four mice in each group began to regain weight from 10 dpi and survived until the observation endpoint, suggesting improvement in disease severity and survival (green and blue lines). Notably, all mice in the IN-prophylaxis group, despite receiving a sixfold lower dose of B5-D3 protein than those in the IV groups, maintained stable body weight and achieved complete survival over the 14-day period (red lines).

To monitor viral burden, one mouse from each group was sacrificed at 4 dpi (*Figure 2A*). Corroborating the body weight and survival data, no infectious viral particles were detected in lung homogenate from the IN-prophylaxis mouse. In contrast, mice treated with IV prophylaxis or therapy showed reduced but still detectable viral titers compared to the PBS group (*Figure 2D*). Immunohistochemistry (IHC) staining further confirmed the absence of viral nucleocapsid (N) protein in the IN-treated mouse, whereas IV-treated mice showed residual infection and immune cell infiltration. H&E staining revealed varying degrees of alveolar thickening in all SARS-CoV-2-inoculated mice (*Figure 2E*).

To further explore the effective timing of IN administration that offers superior antiviral protection, we treated a younger cohort of K18-hACE2 mice (2–3 months old) with B5-D3 (IN, 2.5 mg/kg) at –24, –6, or +24 hr relative to viral challenge (*Figure 2F*). Consistently, all mice in the PBS group of the young cohort exhibited substantial weight reduction from 4 dpi and reached approximately 20% loss by 7 dpi (*Figure 2G*, black lines). Whereas differently from the aged cohort, two of the five infected young mice eventually recovered, resulting in 40% survival (*Figure 2G, H*, black lines) indicating age-related fitness (*Dwivedi et al., 2024*). Interestingly, both the –24 and –6 hr IN-prophylaxis groups maintained stable body weights (*Figure 2G*, pink and red lines), ending up with survival rates of 80% and 100%, respectively (*Figure 2H*). In contrast, the +24 hr IN-therapy group showed substantial weight loss and no survival improvement compared to the PBS group, indicating that high-efficiency antiviral protection of IN B5-D3 is limited to prophylactic administration but not post-infection treatments (*Figure 2G, H*; orange lines). Consistently, virus-neutralizing antibodies were detected in surviving mice from the PBS and +24 hr groups at 14 dpi, indicating once active infection and subsequent immune response. Whereas antibody levels remained minimal in the two IN-prophylaxis groups, suggesting effective prevention of viral replication (*Figure 2I*).

## Efficient protection against SARS-CoV-2 by intranasal B5-D3 prophylaxis depends on Fc-mediated effector functions

Adding on to the 2-week observation of the significant protection conferred by IN prophylaxis with B5-D3, we examined the early responses following SARS-CoV-2 challenge in K18-hACE2 mice. A new cohort of 2- to 3-month-old mice received B5-D3 IN treatment 6 hr before infection (–6 hr), and lung tissues were harvested at 1, 2, and 4 dpi for analysis (*Figure 3A*). An additional group was treated with a modified version of B5-D3, which contains L234A/L235A mutations in the human IgG1 Fc region (B5-D3-LALA) to abolish its binding to Fc gamma receptor (FcγR) and abrogate Fc effector functions (*Lund et al., 1991*; *Figure 3—figure supplement 1*). Quantitative PCR of viral spike (*S*) and nucleocapsid (*N*) RNA in lung tissues revealed only marginal viral loads in the B5-D3-treated mice at as early as 1 dpi, indicating efficient suppression of early viral replication compared to the PBS group (*Figure 3B*). Analysis of infectious viral particles in lung homogenates further corroborated these observations, demonstrating minimal or undetectable viral titers in the B5-D3 group at all time points (*Figure 3C*). In contrast, PBS-treated mice exhibited consistently high viral loads. Interestingly, the analysis of the B5-D3-LALA group detected varied levels of *S* and *N* RNAs and significant viral burdens in two out of three mice, indicating partial protection and supporting that Fc effector functions are

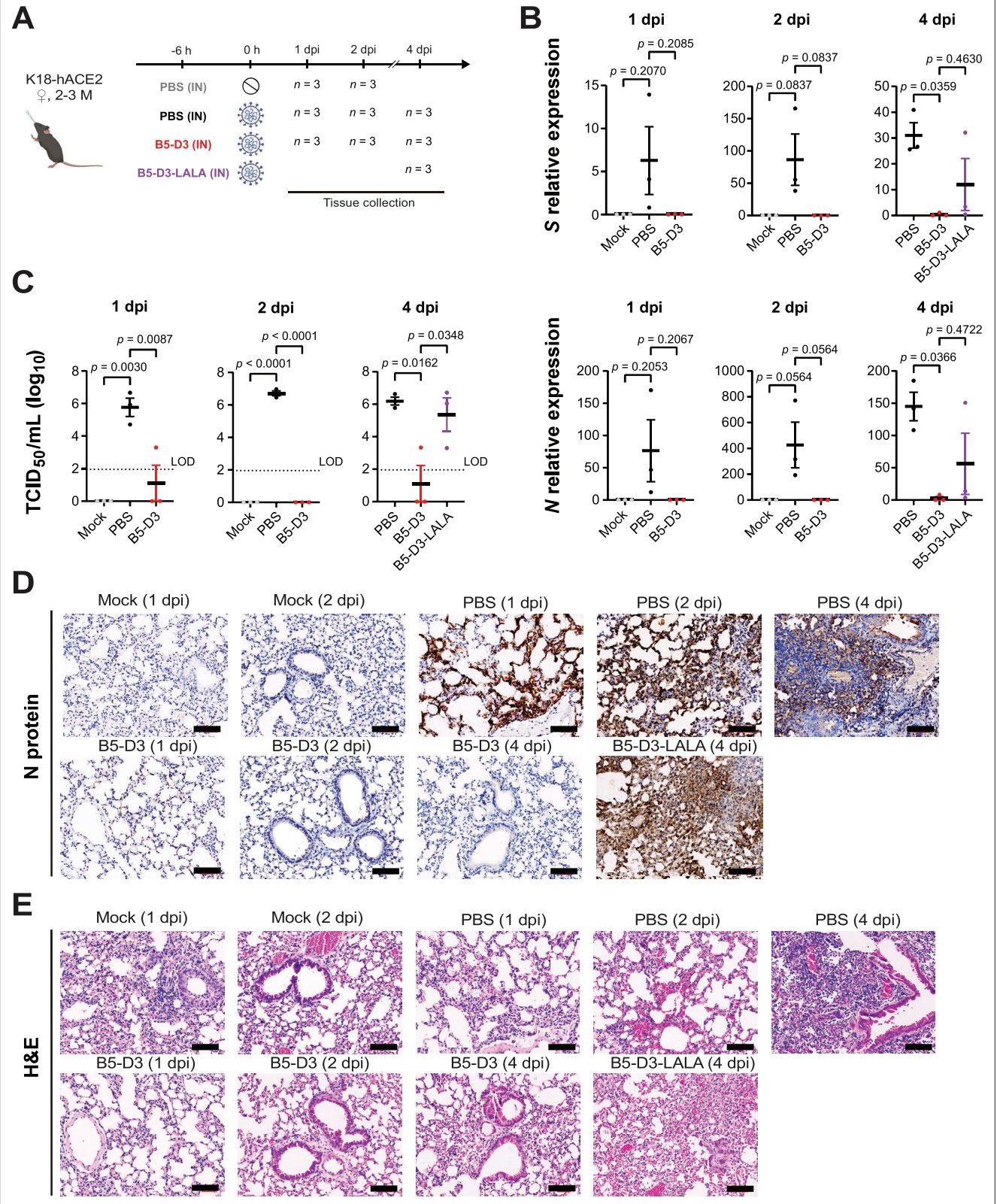

**Figure 3.** Efficient viral clearance at early stages through intranasal prophylaxis with B5-D3 against SARS-CoV-2 challenge in K18-hACE2 mice. (**A**) Workflow diagram showing timelines and treatments for different mouse groups. Young female K18-hACE2 mice aged 2–3 months received prophylactic administration of PBS (black), B5-D3 (red), or B5-D3-LALA (purple) via the IN route 6 hr prior to inoculation with $1 \times 10^4$ PFU of Wuhan-Hu-1. Mice inoculated with PBS instead of the virus served as mock controls (gray). Mice from each treatment group were sacrificed for tissue collection at 1, 2,

*Figure 3 continued on next page*

*Figure 3 continued*

and 4 dpi (n = 3 per time point). (**B**) Quantitative PCR results showing relative amounts of *S* (upper) and *N* (lower) viral RNA in lung tissues collected from different groups at 1, 2, and 4 dpi, normalized to mouse *Gapdh*. (**C**) The titers of infectious viruses detected in lung homogenates, measured by $TCID_{50}$ assays at 1, 2, and 4 dpi. (**D,E**) Fixed lung tissues were sectioned and stained; IHC for viral N protein (**D**) and H&E staining for tissue damage (**E**) are shown (scale bar = 100 μm). h, hour(s). dpi, day(s) post injection. Data presented as mean ± standard error of the mean (SEM). Statistical significance was determined by Tukey's multiple comparisons test.

The online version of this article includes the following figure supplement(s) for figure 3:

**Figure supplement 1.** Comparison between B5-D3 and B5-D3-LALA in in vitro neutralization against the SARS-CoV2 pseudovirus and Fc-mediated effector functions.

**Figure supplement 2.** Entire sections for histological examination at 1 dpi.

**Figure supplement 3.** Thickening of alveolar septum in K18-hACE2 mice after SARS-CoV-2 challenge.

indispensable for full efficacy of IN B5-D3 (*Figure 3B, C*, right panels). Consistently, IHC staining for N protein in lung sections confirmed the absence of viral infection in the B5-D3 groups at all time points. Whereas signs of viral replication were evident in the lungs of mice treated with PBS since as early as 1 dpi and in the B5-D3-LALA-treated cohort as examined at 4 dpi (*Figure 3D*, *Figure 3—figure supplement 2*, left panels). Despite variations in viral burden, H&E staining indicated alveolar septal thickening in all groups (*Figure 3E*, *Figure 3—figure supplement 2*, right panels and *Figure 3—figure supplement 3*). Notably, moderate alveolar thickening persisted in the B5-D3-treated mice till the end point at 4 dpi, whereas the PBS groups developed much severer alveolar thickening at 4 dpi. Consistent with the partial protection observed with B5-D3-LALA, histological analysis of lung samples in this group revealed severer yet heterogeneous alveolar thickening (*Figure 3—figure supplement 3*). These findings collectively demonstrated that IN prophylaxis with B5-D3 blocks SARS-CoV-2 infection not only by neutralization but also by immune mechanisms such as Fc-mediated effector functions.

## RNA-Seq analysis of lung transcriptomes reveals early antigen presentation and prompt viral clearance following SARS-CoV-2 neutralization by B5-D3

To delineate the immune mechanisms underlying IN B5-D3-mediated prophylactic protection against SARS-CoV-2, we examined the transcriptomes of lung samples collected at 1, 2, and 4 dpi from the above experiment (*Figure 4A–D*, *Figure 4—figure supplement 1A*). Unsupervised clustering based on Pearson correlation distinguished samples with severe infection (mainly PBS-treated) from those with subtle or no infection (mocks and most decoy-treated mice) (*Figure 4—figure supplement 1A*). Corroborating the levels of viral infections observed, differential gene expression (DGE) analysis revealed extensive inflammatory responses in the PBS groups, significantly greater than in mock treatments. At 1, 2, and 4 dpi, 26, 1232, and 1756 genes were upregulated, respectively, and were significantly enriched in Gene Ontology Biological Process (GOBP) terms related to antiviral responses such as type I interferon (IFN) responses and innate immune responses (*Figure 4A, B*, *Figure 4—figure supplement 1B–D*; *Winkler et al., 2020*). In stark contrast, DGE analysis between B5-D3 prophylaxis and mocks at 1, 2, and 4 dpi showed subtle changes, with only 1, 7, and 32 genes upregulated, respectively, and only moderate enrichment in chemotaxis-related pathways at 4 dpi (*Figure 4C*, *Figure 4—figure supplement 1E*). The B5-D3-LALA group, however, had 264 genes upregulated at 4 dpi compared to the mocks, suggesting incomplete protection and ongoing viral activity (*Figure 4D*, *Figure 4—figure supplement 1F*).

To capture the immune activations specifically linked to B5-D3-triggered antiviral efficacy other than infection-induced inflammation, we directly compared the B5-D3 and PBS groups (*Figure 4E–J*, *Figure 4—figure supplements 2 and 3*). Interestingly, at 1 dpi, the B5-D3 group exhibited enhanced expression of several immune-related genes, including *Lef1* (*Shan et al., 2021*), *Fscn1* (*Yamakita et al., 2011*), *Kcne4* (*Colomer-Molera et al., 2023*), *Tcrb*, and *Ccl22* (*Rapp et al., 2019*; *Korobova et al., 2023*), which are associated with early dendritic cell function and T cell activation (*Figure 4E*). Gene Set Enrichment Analysis (GSEA) of GOBPs and Kyoto Encyclopedia of Genes and Genomes (KEGG) pathways further supported these findings (*Figure 4F, G*). Chemotaxis and pathways related to antigen presentation such as Rap1 signaling pathway (*Katagiri et al., 2002*) and Th1 and Th2 cell differentiation were significantly activated in the B5-D3 group at 1 dpi compared to PBS group

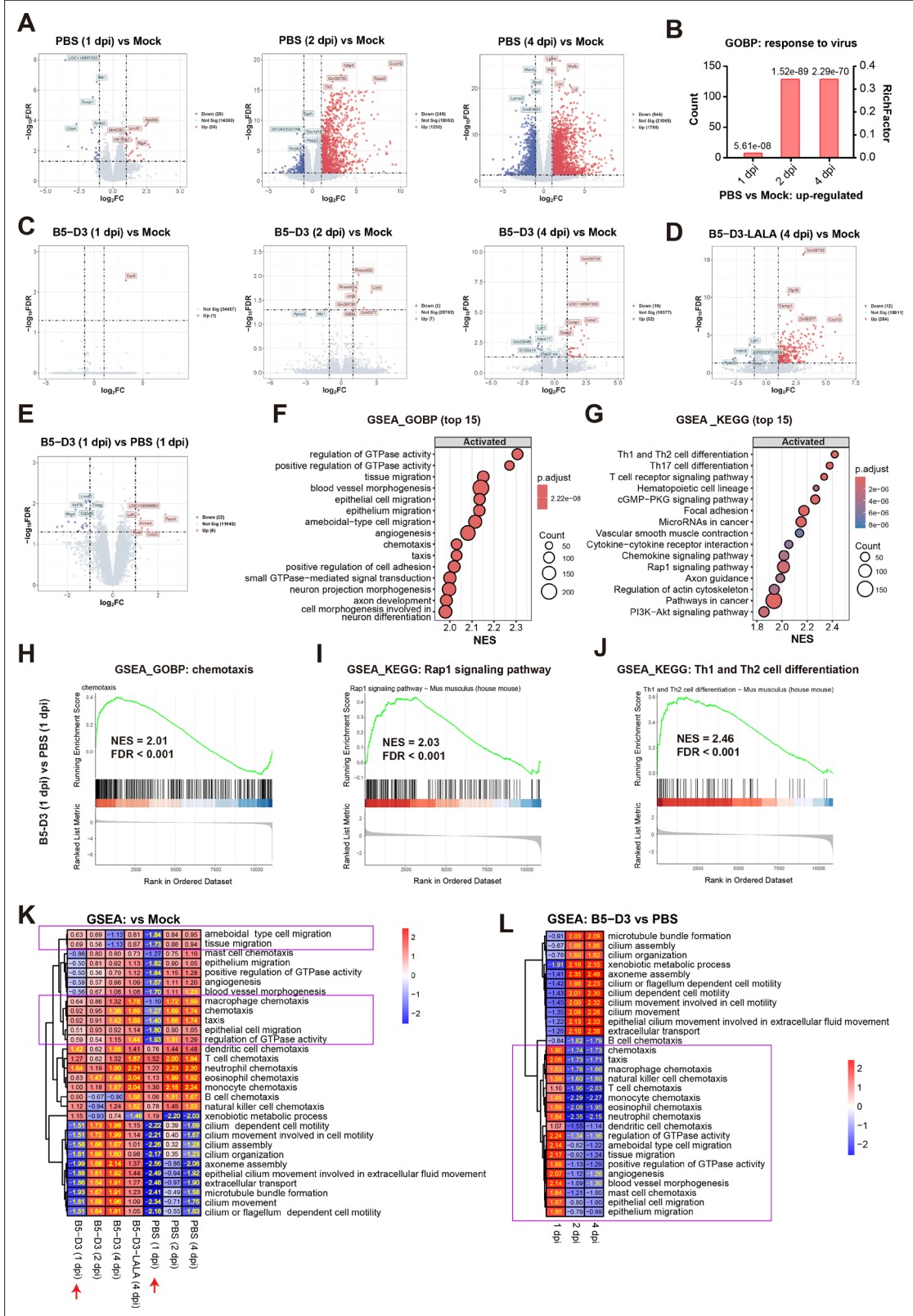

**Figure 4.** Transcriptomic analysis of lungs revealed early immune activation in IN B5-D3-prophylaxis mouse group after SARS-CoV-2 challenge. (**A–D**) DGE analysis comparing PBS (**A**), B5-D3 (**C**), and B5-D3-LALA (**D**) against the mock control at specific time points (n=3). Volcano plots illustrate the gene expression changes (**A, C, D**), while red and blue dots represent significantly upregulated and downregulated genes, respectively, with |log₂ fold change (log₂FC)|≥1 and a false discovery rate (FDR)<0.05. Bar chart in (**B**) shows the enrichment of GOBP 'response to virus' observed in PBS groups

*Figure 4 continued on next page*

*Figure 4 continued*

at 1, 2, and 4 dpi, in which adjusted *p* values are indicated for individual comparisons. (**E–G**) Comparison between IN B5-D3 and PBS group at 1 dpi. Volcano plot illustrates the DGE analysis between IN B5-D3 to PBS group at 1 dpi (**E**), with red and blue dots representing significantly upregulated and downregulated genes, respectively, with |log$_2$FC|≥1 and FDR <0.05. GSEA shows top 15 significantly activated GOBPs (**F**) and KEGG pathways (**G**) in IN B5-D3 compared to PBS group at 1 dpi. NES, normalized enrichment score; p.adjust, adjusted *p* value. (**H–J**) GSEA plots of chemotaxis (**H**), Rap1 signaling pathway (**i**), and Th1 and Th2 cell differentiation (**J**) in B5-D3 vs PBS comparison at 1 dpi. (**K,L**) Heatmaps show NES of GSEA comparing various treatments to the mock control (**K**) and between B5-D3 to PBS (**L**), focusing on top 10 GOBPs in (**F**) and *Figure 4—figure supplement 3C, D*, respectively, and those related to immune cell chemotaxis. Significant NES values (*p* < 0.05, FDR <0.25) are highlighted in yellow. Purple boxes indicate GOBPs where B5-D3 (1 dpi) group shows activation but PBS (1 dpi) group shows suppression. Benjamin–Hochberg method was used for FDR adjustment.

The online version of this article includes the following figure supplement(s) for figure 4:

**Figure supplement 1.** RNA-Seq analysis of K18-hACE2 mouse lungs with different pretreatments upon SARS-CoV-2 challenge.

**Figure supplement 2.** Leading-edge subsets in GSEA.

**Figure supplement 3.** Transcriptomic comparisons between B5-D3 and PBS pretreatments in K18-hACE2 mice upon SARS-CoV-2 challenge.

**Figure supplement 4.** Minimal transcriptomic alterations in lungs after IN B5-D3 administration without viral challenge.

(*Figure 4H–J*, *Figure 4—figure supplement 2A–C*). Moreover, the B5-D3 groups showed enhancement in cilium movement and metabolism of xenobiotics at both 2 and 4 dpi, suggesting active clearance of viral particles due to effective early responses (*Figure 4—figure supplement 3C–F*).

Furthermore, we collectively examined the GOBPs that were significantly activated in B5-D3 groups at either 1, 2, or 4 dpi among all treatment groups and time points. Markedly, the B5-D3 group showed higher normalized enrichment scores (NES) in chemotaxis-related GOBP pathways than the PBS group at 1 dpi (*Figure 4K*, purple boxes), while direct comparison between B5-D3 and PBS groups further revealed the broad involvement of multiple types of effector immune cells (*Figure 4L*, purple box). These results collectively indicate that early immune activation is a hallmark of B5-D3-mediated protection.

Finally, the lung transcriptomes from mice receiving B5-D3 without viral inoculation showed high similarity to the PBS vehicle controls (*Figure 4—figure supplement 4A*). The 10 upregulated genes identified showed poor correlation with the virus-inoculated B5-D3 group (*Figure 4—figure supplement 4B, C*), supporting that early immune responses observed in B5-D3 IN prophylaxis groups were primarily triggered by virus neutralization rather than by B5-D3 alone.

## Intranasally delivered B5-D3 is enriched in the respiratory tract and targets mainly the airway macrophages

The superior prophylactic antiviral effects of IN over IV administration of B5-D3, as observed in the K18-hACE2 infection experiments (*Figure 2A–E*), suggested the importance of mobilizing the local immunity within the respiratory tract. Next, we labeled B5-D3 protein with Alexa Fluor 750 (AF750) and examined its bio-distribution and kinetics after IN administration (*Figure 5A*). In vivo imaging showed that fluorescence-labeled B5-D3 (B5-D3-AF750) was present in the nasal cavities for at least 24 hr after a single IN dose in K18-hACE2 mice (*Figure 5B*). Ex vivo images further revealed that B5-D3-AF750 distributed in the respiratory tract from nasal cavity to lung within 20 min and remained enriched in lungs by 24 hr after administration (*Figure 5C*). In contrast, non-respiratory organs showed minimal signals, which were merely detectable in the urinary system and liver (*Figure 5D*).

To identify the immune cells in the respiratory tract that are actively engaged with IN B5-D3, we performed flow cytometry analysis on bronchoalveolar lavage fluid (BALF) at 6 hr after IN administration of B5-D3-AF750 (*Figure 5E*). Like normal conditions, over 95% of live BALF cells were CD45$^+$ immune cells, predominantly composed of CD11c$^+$Siglec-F$^+$ resident alveolar macrophages (AMs) (>50%) in both treatment and vehicle groups (*Figure 5F*, *Figure 5—figure supplement 1*). Notably, the IN administered B5-D3-AF750 was actively retained in the CD45$^+$ cells, with positive rates exceeding 65% in all treated mice (*Figure 5G*). Among the CD45$^+$B5-D3$^+$ cells, more than 95% were macrophages, composed primarily of CD11c$^+$Siglec-F$^+$ AMs (87.2–91.7%) and Siglec-F$^-$CD11b$^-$F4/80$^+$ monocyte-derived macrophages (mono-Macs; 6.6–9.9%) (*Figure 5H*, red arrows). Consistently, these macrophage populations also exhibited the highest B5-D3 positive rates (*Figure 5I, J*, *Figure 5—figure supplement 2*) and greatest median fluorescent intensities (MFI) (*Figure 5K*, red arrows) among

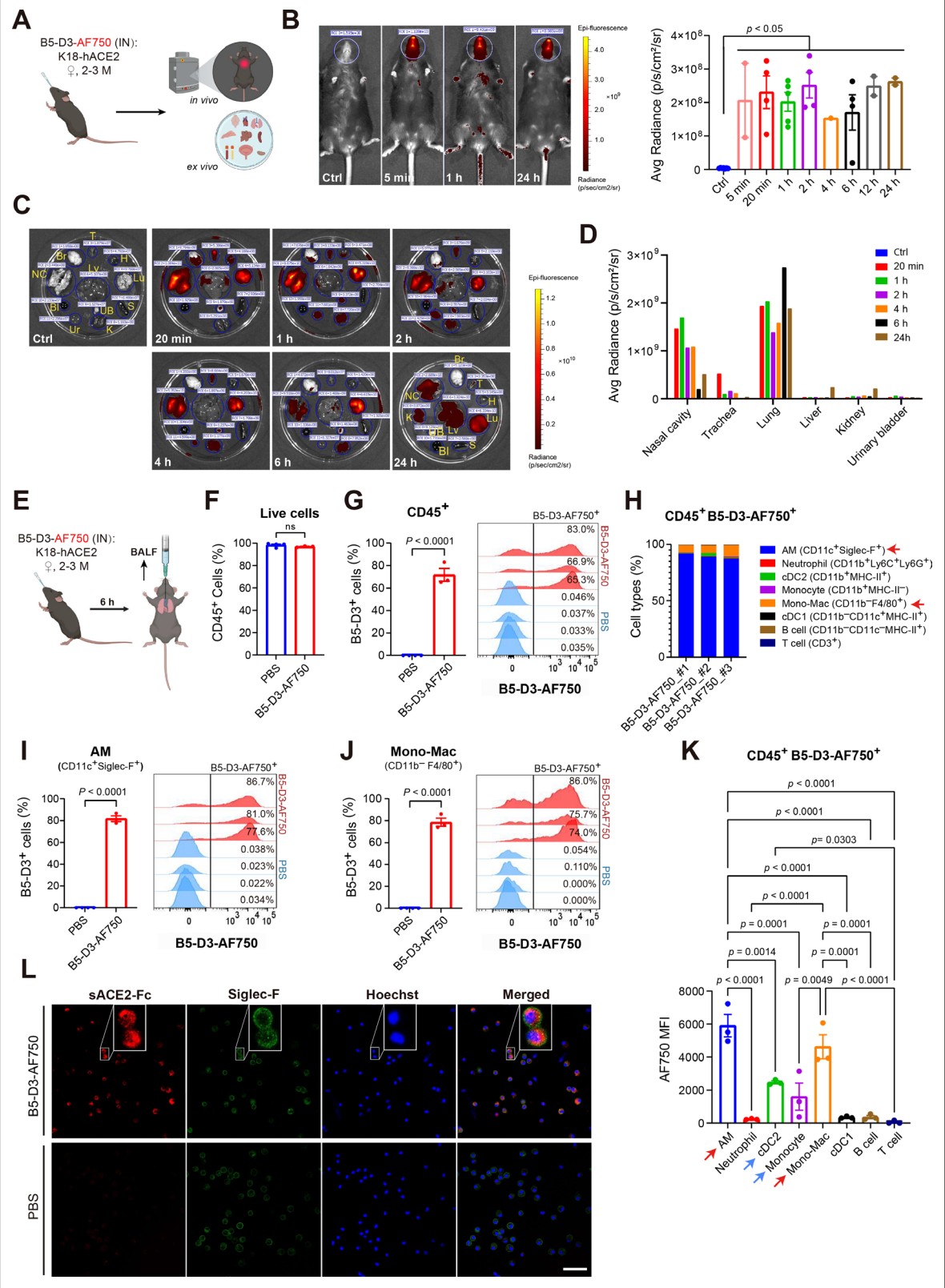

**Figure 5.** In vivo bio-distribution of B5-D3 after IN administration. (**A**) Schematic workflow of in vivo and ex vivo imaging. Female K18-hACE2 mice aged 2–3 months received IN administration of fluorescently labeled B5-D3 (B5-D3-AF750) and were visualized at different time points. (**B**) Representative whole-body images of control and treated mice at 5 min, 1 hr, and 24 hr after B5-D3-AF750 administration, showing the signal captured by in vivo imaging (left). White circles indicate regions of interest (ROIs) for quantification of fluorescence signals in the nasal cavities. Average (Avg) Radiance

*Figure 5 continued on next page*

*Figure 5 continued*

measured at all time points is shown on the right. (**C**) Ex vivo images of tissues from control and treated mice sacrificed at indicated time points after B5-D3-AF750 administration. Blue circles indicate ROIs for signal quantification. Br, brain; NC, nasal cavity; T, trachea; Lu, lung; H, heart; Lv, liver; S, spleen; K, kidney; UB, urinary bladder; Bl, blood; Ur, urine. (**D**) Avg Radiance shows the fluorescence signals in excised tissues measured ex vivo. (**E**) Schematic workflow for BALF analysis. Female K18-hACE2 mice aged 2–3 months received IN administration of B5-D3-AF750 (n = 3) or PBS (n = 4) and were sacrificed 6 hr later for collection of BALF cells. (**F**) Percentage of CD45$^+$ cells in live BALF cells. (**G**) Positive rates (left) and histograms (right) of B5-D3 binding/uptake in CD45$^+$ BALF cells. Histograms show B5-D3-AF750 fluorescence intensities in CD45$^+$ BALF cells from individual mice. (**H**) Frequency of individual immune cell types in CD45$^+$B5-D3$^+$ BALF cells. Red arrows point out AMs and mono-Macs with high abundance. AM, alveolar macrophage; Mono-Mac, monocyte-derived macrophage; cDC1/2, type 1 or 2 conventional dendritic cells. (**I,J**) Positive rates (left) and histograms (right) of B5-D3 binding/uptake in CD11c$^+$Siglec-F$^+$ AMs (**i**) and CD11b$^-$F4/80$^+$ mono-Macs (**J**). (**K**) Median fluorescence intensity (MFI) of AF750 indicates B5-D3 binding/uptake in different CD45$^+$B5-D3$^+$ populations. (**L**) Confocal images (scale bar = 50 μm) of BALF cells collected at 6 hr and stained for sACE2-Fc (red, anti-Fc, Abcam #ab98596), Siglec-F (green, BD #564514), and nuclei (blue, Hoechst). Magnified views are shown in white rectangles. h, hour(s). Data are presented as mean ± SEM, and statistical significance was determined by Tukey's multiple comparisons test or Student's *t*-test.

The online version of this article includes the following figure supplement(s) for figure 5:

**Figure supplement 1.** Flow cytometry analysis of mouse BALF cells.

**Figure supplement 2.** Binding/uptake rates of B5-D3-AF750 in BALF cells.

all immune cell types in the BALF, indicating the strongest B5-D3-AF750 uptake. Other phagocytic cell types such as the type 2 conventional dendritic cells (cDC2) and monocytes also exhibited considerable AF750 intensities (*Figure 5K*, blue arrows; *Figure 5—figure supplement 2B, C*), suggesting potential relationships between B5-D3 uptake and phagocytic activities. Confocal microscopy of BALF cells after immunostaining further confirmed that the B5-D3-AF750 were present in the cytoplasm after being retained in AMs (*Figure 5L*). These results demonstrate that IN B5-D3 preferentially accumulates in the respiratory tract and is predominantly taken up by airway macrophages, supporting their important role in mediating early immune responses.

## sACE2-Fc facilitates phagocytosis of SARS-CoV-2 pseudovirus via mechanisms distinct from ACE2-dependent viral infection

To examine the implication of macrophage involvement in the early immune activation observed in IN B5-D3 treatment groups, we performed cellular analysis using THP-1 cells as an in vitro model for phagocytes (*Chanput et al., 2014*) and examined the sACE2-Fc-dependent phagocytosis of spike-pseudotyped lentiviruses. Indeed, immunostaining of HIV capsid protein p24 confirmed the attachment and entry of pseudoviruses in the THP-1 cells in a B5-D3-dependent manner, with an evidenced signal peak at 6 hr post-co-incubation (*Figure 6—figure supplement 1*). Interestingly, analysis on the THP-1-derived M0 and M1 macrophages detected even greater p24 signals, indicating stronger phagocytosis activities compared to undifferentiated THP-1 cells (*Figure 6A, B*, *Figure 6—figure supplement 2A, B, D, E*). This process resembled antibody-dependent cellular phagocytosis (ADCP), which is significant in THP-1-derived M0 and M1 macrophages (*Tedesco et al., 2018*). Consistently, further examination revealed colocalization of internalized pseudovirus with lysosomal associated membrane protein 1 (LAMP1), indicating trafficking to lysosomes potentially for degradation (*Takeda and Akira, 2005*; *Figure 6C*, *Figure 6—figure supplement 2C, F*).

We further examined the pseudovirus uptake in Calu-3 cells overexpressing human ACE2 (hACE2-Calu-3; *Figure 6—figure supplement 3*) as a model of lung epithelial cells. In contrast to that in macrophages, co-incubation with B5-D3 significantly reduced the pseudovirus entry in hACE2-Calu-3 cells (*Figure 6D–F*). Interestingly, while the pseudovirus transduction in hACE2-Calu-3 cells, in the absence of B5-D3, produced robust luciferase signal indicating viral genome release after cell entry, the evident pseudovirus uptake facilitated by B5-D3 in the THP-1 and derivative macrophages yielded no detectable luciferase activity, which further supported viral degradation within phagolysosomes (*Figure 6G*). Corroborating these observations, western blot analysis showed absence of cleaved S2′ fragments in the macrophages that had internalized pseudovirus-B5-D3 complexes, supporting that the pseudoviruses did not undergo membrane fusion or cytosolic release as it did in the epithelial cell model (*Yu et al., 2022*; *Figure 6H*).

To evaluate the contribution of Fc-mediated effector functions to virus uptake in macrophages, we further compared the efficiencies of B5-D3, B5-D3-LALA, and hIgG1 isotype in mediating pseudovirus uptake by THP-1-derived macrophages (*Figure 6—figure supplement 4*). As indicated by p24

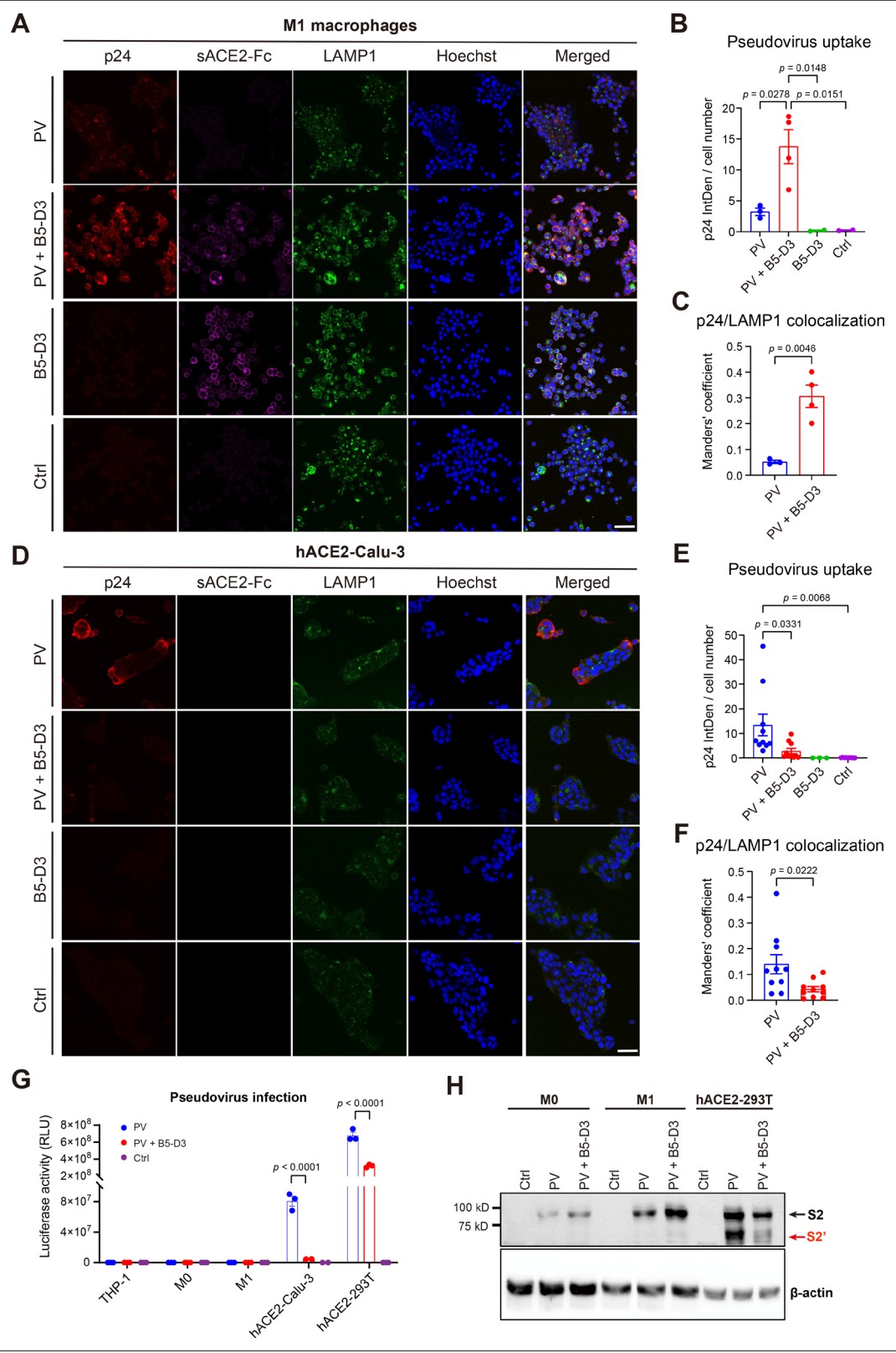

**Figure 6.** B5-D3 enhanced phagocytosis and degradation of SARS-CoV-2 pseudovirus in THP-1-derived macrophages. (**A**) Immunostaining of p24 (Invitrogen #PA5-81773), sACE2-Fc, and LAMP1 (Abcam #ab25630) in THP-1-differentiated M1 macrophages showing phagocytosis of SARS-CoV-2 pseudovirus (PV, p24+) after 6 hr of incubation with or without B5-D3 (scale bar = 50 μm). LAMP1 was stained to identify lysosomes. (**B**) Quantification

*Figure 6 continued on next page*

*Figure 6 continued*

of p24 signal intensity as shown in (**A**). Intensity Density (IntDen) per cell number indicates the mean p24 signal per cell, calculated using ImageJ. Each dot represents one image. (**C**) Manders' coefficient indicating the colocalization of p24 and LAMP1 in THP-1 M1 macrophages as shown in (**A**). (**D**) Immunostaining of p24, sACE2-Fc, and LAMP1 in hACE2-Calu-3 cells after 6 hr incubation with pseudovirus, with or without B5-D3 (scale bar = 50 μm). (**E**) Quantification of mean p24 signal intensity as shown in **D**. (**F**) Manders' coefficient for the colocalization of p24 and LAMP1 in hACE2-Calu-3 cells, as shown in D. (**G**) Quantification of pseudovirus infection in THP-1, M0 macrophages, M1 macrophages, hACE2-Calu-3, and hACE2-293T cells, in the presence or absence of B5-D3. Results shown are luciferase activities measured at 2 days post-transduction. (**H**) Immunoblot staining of cell lysates to detect SARS-CoV-2 spike cleavage after cell entry. M0 macrophages, M1 macrophages, and hACE2-293T cells were incubated with pseudovirus for 6 hr, with or without B5-D3, before protein extraction. Band locations of SARS-CoV-2 spike S2 and S2′ fragments are labeled in black and red, respectively. Data are presented as mean ± SEM, and statistical significance was determined by Tukey's multiple comparisons test.

The online version of this article includes the following source data and figure supplement(s) for figure 6:

**Source data 1.** The original files of the full raw uncropped, unedited blots for SARS-CoV-2 spike S2 and S2′ and β-actin.

**Source data 2.** The original files of the blots for SARS-CoV-2 spike S2 and S2′ and β-actin with relevant bands labelled.

**Figure supplement 1.** Time-course analysis of sACE2-Fc-dependent pseudovirus entry in THP-1 cells.

**Figure supplement 2.** Enhanced phagocytosis of SARS-CoV-2 pseudovirus by THP-1 and THP-1-derived macrophages facilitated by sACE2-Fc.

**Figure supplement 3.** Generation of Calu-3 cell overexpressing hACE2 for enhanced pseudoviral infection.

**Figure supplement 4.** Reduced uptake of SARS-CoV-2 pseudovirus in THP-1-derived macrophages due to malfunction or absence of Fc domain in B5-D3.

**Figure supplement 5.** Transcriptomic analysis revealed activation of THP-1-derived macrophages mediated by 6 hr incubation with B5-D3-pseudovirus complex.

---

staining, functional impairment (LALA mutations) of Fc in B5-D3 significantly reduced the efficiency of virus uptake, emphasizing the importance of intact Fc in B5-D3 function as decoy. Whereas the presence of hIgG1 isotype control showed no impact on pseudovirus uptake, supporting that the phagocytosis was largely specific to the pseudovirus-decoy complexes.

To further evaluate the responses triggered by the pseudovirus–B5-D3 complex in macrophages, we performed RNA-Seq analysis. The transcriptomic profiling revealed little difference and identified no DEG between the macrophages incubated with both pseudovirus and B5-D3 and those treated with pseudovirus only (*Figure 6—figure supplement 5A*). Whereas interestingly, GSEA detected broad activation of pathways related to antiviral response and macrophage activation in the macrophages internalizing pseudovirus and B5-D3 (*Figure 6—figure supplement 5B–I*), corresponding to ADCP effect. These findings indicate moderate immune activation triggered by phagocytosis of pseudovirus-decoy complexes, corroborating with the mild immune activation that accelerated antiviral responses in our mice infection experiments.

Collectively, these findings suggest that IN B5-D3 not only blocks viral entry into epithelial cells but also actively redirects SARS-CoV-2 to phagocytic clearance by engaging airway phagocytes via Fc-dependent mechanisms. Moreover, such ADCP-like process contributes to early immune activation and restricts the infection at the respiratory mucosal surface.

## Discussion

In this study, we comprehensively evaluated the protective efficacy and mechanistic basis of an optimized representative sACE2-Fc decoy (B5-D3) against SARS-CoV-2 infection. By introducing only two mutations (T92Q and H374N), we generated a minimally engineered sACE2-Fc mutant (B5-D3) that achieved broad-spectrum neutralization with minimal risk of disrupting the RAS. Among various administration routes and dosing schedules examined, we demonstrated that IN prophylaxis of B5-D3 achieved the most robust protection, completely preventing disease in both young and aged K18-hACE2 mice. Transcriptomic analysis of the infected lung samples at early time points revealed distinct IN B5-D3-dependent immune activation at the onset of infection, indicating that B5-D3 acted not only

as a receptor decoy but also as an immune engager upon viral infection. Bio-distribution analysis of fluorescence-labeled B5-D3 demonstrated rapid uptake and high accumulation in the respiratory tract, primarily within airway macrophages. Phagocytosis assays further supported that sACE2-Fc decoy mediated a rapid viral clearance in macrophages, while abolishing membrane ACE2-mediated infection in epithelial cells. Together, these findings reveal a dual-function mechanism for sACE2-Fc decoys in redirecting SARS-CoV-2 to phagocytic clearance and rapid immune engagement, supporting their potential as intranasal prophylactics against respiratory viruses.

Previous studies have reported multiple ACE2 mutations with remarkable potential in neutralizing SARS-CoV-2. However, in these studies, combining ACE2 mutations based on in silico predictions to both enhance spike binding and eliminate the ACE2 enzymatic activity in a single design resulted in accumulation of mutations such as K31F/N33D/H34S/E35Q/H345L (*Glasgow et al., 2020*) and L79F/M82Y/Q325Y/H374A/H378A (*Chen et al., 2022*). These extensive mutations have been implicated in structural instability (*Glasgow et al., 2020*) and reduced production efficiency (*Chan et al., 2020*). More importantly, the high mutation loads raise risks for immunogenicity, which is a critical issue when considering clinical applications. Corroboratively, Urano et al. detected in vitro T cell stimulation elicited by the L79F mutation, whereas the T92Q mutation showed much lower immunogenicity and enhanced spike binding affinity (*Urano et al., 2023*). In our ACE2 decoy design, we incorporated only two mutations (like T92Q and H374N in B5-D3) to enhance neutralization potency while eliminating enzymatic activity, resulting in simplest compound ACE2 mutants. The B5-D3, as a representative, exhibited not only minimal mutation-related risks but also top-level neutralization potencies among all candidate mutants we tested. Hence, by coupling structural engineering (Fc fusion for avidity improvement) and mutagenesis (potency and safety optimization), we provided simplest decoy design and further consolidated the generalizable framework for decoy design, which is invaluable to combat a novel, highly contagious, and fatal virus in future, before more effective vaccines and drugs are established.

Previous studies have reported that IN prophylaxis with sACE2-Fc mutants or monoclonal antibodies (*Hassler et al., 2023*; *Kober et al., 2024*; *Ku et al., 2021*) could effectively protect against SARS-CoV-2, in which the Fc-effector functions were indispensable (*Chen et al., 2022*). However, the precise mechanism has not been well depicted, which prevents the further development of these

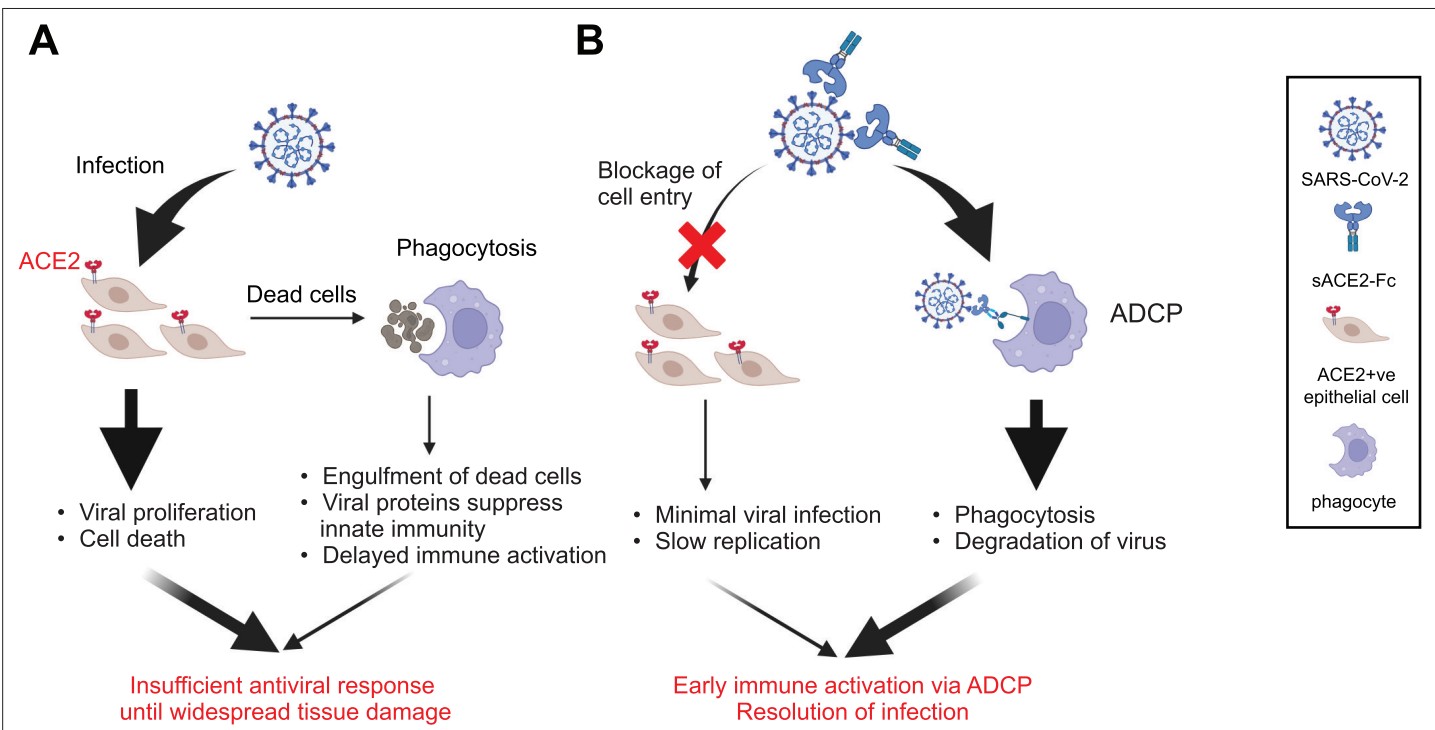

**Figure 7.** Proposed mechanisms of action of IN sACE2-Fc decoy in preventing SARS-CoV-2 infection. Schematics illustrating the actions and outcomes of SARS-CoV-2 infection, in the absence (**A**) and presence (**B**) of IN delivered sACE2-Fc decoys. The figure was created in BioRender.com.

approaches for translation. Here, our study provided evidence for a deeper mechanistic insight, showing that IN sACE2-Fc decoys rapidly engage host immunity in the respiratory tract. Consistent with others' work (*Chen et al., 2022*), IN delivery of an Fc-null variant (B5-D3-LALA) resulted in suboptimal protection and higher viral infection compared to B5-D3. Notably, despite the minimal infection observed, B5-D3-treated mice showed robust early immune activation, including induction of antigen presentation and T cell activation within 24 hr post-infection (*Figure 4*). These results support that B5-D3 not only neutralizes virus but also primes innate and adaptive immune responses, counteracting early-stage viral immune evasion.

Furthermore, our bio-distribution data showed that IN-delivered B5-D3 preferentially accumulates in the respiratory tract (*Figure 5A–D*). Flow cytometry and confocal imaging confirmed strong binding and uptake of B5-D3 by airway phagocytes, primarily AMs and monocyte-derived macrophages (*Figure 5E–L*). Notably, phagocytosis assays demonstrated that B5-D3–virus complexes were trafficked to lysosomes for degradation in macrophages. Functional impairment in the B5-D3 structure reduced the uptake of virus by macrophages. These findings support a mechanism in which B5-D3 redirects viral particles away from membrane ACE2-dependent epithelial entry and toward phagocytic clearance (*Figure 7*). Importantly, RNA-Seq analysis of macrophages treated with virus-decoy complexes suggests that such ADCP-like processes also facilitated early immune activation and initiated downstream antiviral signaling cascades before the virus reaches epithelial targets (*Figure 7B*). Hence, the IN prophylaxis offers a unique advantage by enabling localized immune priming and efficient viral clearance at the frontline of infection. The IN-administered antibody drugs can share a similar mechanism and confer effective protection against primary infection of SARS-CoV-2 or other respiratory viruses. However, antibody drugs can be easily escaped when new viral variants emerge, which is specifically common among rapidly spreading respiratory RNA viruses (*Carabelli et al., 2023*). In contrast, the decoy strategy is intrinsically resistant to 'antigenic escape' of the mutating virus, which is further supported by the trend observed in viral evolution that later-emerging SARS-CoV-2 variants exhibit a higher affinity for the ACE2 receptor, enhancing their infectivity and transmissibility (*Kober et al., 2024*).

In a post-infection context, respiratory viruses would have propagated extensively at the primary infection site when symptoms arise, which makes IN neutralization relatively ineffective to clear the high viral burden and the resultant systemic inflammation (*Cheemarla et al., 2021*). Therefore, the therapeutic activity of either local or systemic neutralization at the post-infection stage is bound to be restricted, which is intrinsically inherited in all decoy/antibody therapies against respiratory viral infections. Yet, transfusion of convalescent plasma or infusion of neutralizing mAbs in clinical practice have been shown to reduce the mortality in COVID-19 (*Senefeld et al., 2023*; *Gupta et al., 2021*), and IV administration of B5-D3 as therapy was also shown effective in improving the viral burden and survival in our aged mouse experiment (*Figure 2A–E*). Therefore, the therapeutic value of our decoy strategy does exist when a 'systematic' route of administration (e.g., IV infusion) is chosen, and improvement in disease severity should thus be expected.

Our study has several limitations. First, we didn't examine the potential of IgM-based sACE2 decamers, which showed higher avidity to spikes and greater potency for viral neutralization in previous studies (*Ku et al., 2021*; *Liu et al., 2023*; *Guo et al., 2023*). It is promising that our B5-D3 design would benefit from switching to the IgM isotype, whereas the distinct biological features imposed by IgM Fc, including short serum half-life and restricted tissue penetration (*Keyt et al., 2020*), may complicate the analysis design and diverge our study focus. Moreover, although no sign of B5-D3-specific immune responses was observed in our AAV-injected mice, its immunogenicity and potential risk for antibody-dependent enhancement should be thoroughly evaluated to further develop the decoy strategy for human use. Furthermore, despite that K18-hACE2 mice are broadly used and demonstrate the best susceptibility to SARS-CoV-2 infection among established hACE2-transgenic mouse models (*Lutz et al., 2020*), hACE2 expression in these mice shows a distinct pattern that may not reflect human physiology (*McCray et al., 2007*; *Oladunni et al., 2020*). Further study on hACE2 decoy using animals with physiological levels of hACE2 expression and a humanized immune system that can support stable engraftment of myeloid lineages would be warranted.

In conclusion, we present a rationally designed sACE2-Fc decoy with minimal mutagenesis (B5-D3) and provide compelling evidence and insights into the immune mechanism supporting its potent prophylactic efficacy. Intranasal prophylactic administration of B5-D3 not only neutralizes SARS-CoV-2

but also redirects the virus toward phagocytic clearance, enabling early immune engagement and complete protection. These findings provide a mechanistic basis for decoy-based antiviral strategies and offer a promising approach to combat current and future airborne viral threats. Further studies may aim to develop approaches to enhance the rapid local immune engagement to restrict early viral propagation. Additionally, regimen refinements are needed to enhance the stability and functionality of decoy-based treatments before their clinical translation and extension to a broader range of respiratory pathogens.

## Materials and methods

### Plasmid construction
The coding sequence of human ACE2 was cloned into the pGEM-T easy vector (Promega) and underwent site-directed mutagenesis (*Payandeh et al., 2020*; *Lei et al., 2020*; *Glasgow et al., 2020*; *Chan et al., 2020*). The sACE2 and human IgG1 hinge-Fc regions (aa 216–447) were assembled via overlapping PCR. These constructs, along with 6xHis-tagged versions, were inserted into the HDM-SARS2-Spike-delta21 vector (Addgene #155130) to generate HDM-CMV-sACE2(-Fc)-his plasmids. L234A/L235A (LALA) in hIgG1 were introduced to generate HDM-CMV-sACE2-Fc-LALA-his plasmids. Various sACE2-Fc fragments were then subcloned into pAAV-nEFCas9 vector (Addgene #87115) to generate AAV-nEF-sACE2-Fc plasmids. SARS-CoV-2 spike variants with or without an HA tag fused to the C-terminal were synthesized and inserted into the HDM vector (*Shu and McCauley, 2017*).

### Protein structure visualization
The crystal structure of SARS-CoV-2 spike receptor-binding domain bound with ACE2 (6M0J) was downloaded from the Protein Data Bank (PDB, https://www.rcsb.org/structure/6m0j) (*Lan et al., 2020*). Color-labeling of individual amino acids was performed on the PDB website. For structural overlapping analysis of WT sACE2 and B5-D3 (aa 18–740), protein structures were predicted using the online AlphaFold 3 server (https://alphafoldserver.com/) (*Abramson et al., 2024*). PyMOL was utilized for RMSD calculations and structural visualization.

### Cell culture
293T (American Type Culture Collection, ATCC #CRL-3216), Vero E6 (ATCC #CRL-1586), Calu-3 (ATCC #HTB-55), and THP-1 (ATCC #TIB-202) cells, incubated at 37°C with 5% $CO_2$, were authenticated by STR profiling and tested negative for mycoplasma. Specifically, 293T, Vero E6, and Calu-3 cells were maintained in Dulbecco's Modified Eagle Medium (DMEM, Gibco #11965092) supplemented with 10% fetal bovine serum (FBS, Gibco # A5256501) and 1% penicillin–streptomycin (PS, Gibco #15140148). THP-1 cells were cultured in Roswell Park Memorial Institute 1640 medium (RPMI, Gibco #11875093) with similar supplements. THP-1 cells were differentiated into M0 macrophages using 50 nM phorbol 12-myristate 13-acetate (PMA) for 48 hr, followed by a 24-hr rest. For M1 macrophage differentiation, post-PMA treatment cells were stimulated with 10 ng/ml lipopolysaccharide and 20 ng/ml IFN-γ for 24 hr. Expi293F cells (Gibco # A14635) were cultured following the manufacturer's instructions.

### Immunofluorescence staining of hACE2-293T and hACE2-Calu-3
Cells were fixed, permeabilized, and blocked with 10% Normal Goat Serum (Invitrogen). ACE2 was stained with a primary antibody (Abcam #ab15348) followed by an Alexa Fluor 594-conjugated secondary antibody (Invitrogen #A-21442). Cells were counterstained with Hoechst 33342 (Thermo Scientific) and examined under a Nikon Ti2-E Inverted Fluorescence Microscope.

### Lentivirus packaging and transduction
293T cells were seeded at 80% confluence and transfected with psPAX2 (Addgene #12260), pMD2.G (Addgene #12259), and transfer plasmid pWPI-IRES-Puro-Ak-ACE2-TMPRSS2 (Addgene #154987) using polyethylenimine (PEI). Lentivirus-containing medium was harvested 72 hr post-transfection, filtered through a 0.45-µM filter, concentrated, and stored at –80°C. For transduction, 293T or Calu-3 cells were exposed to the concentrated lentivirus with 8 µg/ml polybrene for 24 hr to obtain human ACE2-overexpressing cell lines (hACE2-293T and hACE2-Calu-3, respectively).

## Pseudovirus packaging, titration, and infection

Pseudoviruses were packaged in 293T cells using pCDH-EF1a-eFFly-eGFP (Addgene #104834) and spike-encoding plasmids, following a similar protocol to that of lentivirus. Post-packaging, pseudo-viral particles were titrated using the Lenti-X qRT-PCR Titration Kit (Takara #631235) and used to infect target cells in the presence of 8 µg/ml polybrene. Infectivity was assessed via a luciferase assay (Promega #E1501).

## Protein production and purification

293T and Expi293F cells were transfected with HDM-CMV-sACE2(-Fc)(-LALA)-his plasmids using PEI and ExpiFectamine 293 Transfection Kit (Gibco), respectively. Culture supernatants were collected after 72 hr and 5 days post-transfection, respectively. The 293T supernatant was assessed for ACE2 and IgG1 levels using enzyme-linked immunosorbent assay (ELISA) kits (Abcam #ab235649; Invitrogen #BMS2092), and ACE2 activity was measured with a fluorometric assay (Abcam # ab273297). The Expi293F supernatant underwent Ni-NTA Agarose purification, followed by elution and buffer exchange to phosphate-buffered saline (PBS, pH 7.4). Protein concentration and integrity were verified using the Bradford method, ELISA, and sodium dodecyl sulfate–polyacrylamide gel electrophoresis (SDS–PAGE).

## In vitro pseudovirus neutralization assay

Conditioned media containing sACE2, sACE2-Fc, or sACE2-Fc-LALA proteins were diluted serially, mixed with pseudovirus ($4 \times 10^9$ copies), and incubated at room temperature for 30 min (*Crawford et al., 2020*). The mixture was added to hACE2-293T cells in 96-well plates with duplicates with 8 µg/ml polybrene. Transduction efficiency was assessed 48 hr later via green fluorescent protein (GFP) imaging and/or luciferase assays.

## Reporter-based in vitro ADCC and ADCP assays

The in vitro antibody-dependent cell-mediated cytotoxicity (ADCC) and ADCP activities of B5-D3(-LALA) were measured using Jurkat-Lucia NFAT-CD16 and Jurkat-Lucia NFAT-CD32 cells (InvivoGen), respectively, according to the manufacturer's instructions. 293T cells transfected with pBOB-CAG-SARS-CoV-2-Spike-HA (Addgene #141347) acted as target cells. Target cells were co-incubated with reporter cells and serially diluted B5-D3(-LALA) at 37°C for 1 hr. Luciferase expression indicating CD16 and CD32 signaling was measured using QUANTI-Luc (InvivoGen).

## ADCP of pseudovirus in THP-1 and THP-1-derived macrophages

SARS-CoV-2 pseudovirus (Wuhan-Hu-1, $8 \times 10^8$ copies in 10 µl) was mixed with 50 µl of B5-D3 or control proteins (20 µg/ml) and added to THP-1, M0, or M1 cells cultured in a µSlide 18 Well iBITreat chamber slide. Here THP-1 cells were attached on the collagen-coated iBITreat chamber slide. After 1, 3, 6, or 18 hr of incubation for phagocytosis, cells were fixed with ice-cold methanol for 10 min, blocked with 10% normal goat serum for 1 hr at room temperature, and immunostained sequentially for human IgG-Fc (Abcam #ab98596), HIV-1 p24 (Invitrogen #PA5-81773), and lysosomal associated membrane protein 1 (LAMP1) (Abcam #ab25630). Secondary antibodies were applied (Invitrogen #A-21200 and #A-31573), and nuclei were stained with Hoechst. Imaging was conducted using a Leica SP8 confocal microscope. p24 immunofluorescence intensity and colocalization with LAMP1 (Manders' coefficient) were analyzed using ImageJ and the JACoP plugin.

For RNA extraction, M0 cells were washed once with PBS and added with TRIzol (Invitrogen) after 6-hr incubation with pseudovirus with/without B5-D3 (*n* = 3 in triplicate wells).

## Western blot for spike cleavage detection

SARS-CoV-2 spike-HA tagged pseudovirus ($4 \times 10^9$ copies) was incubated with M0/M1 macrophages or hACE2-293T cells for 6 hr, with or without sACE2-Fc proteins. After incubation, cell lysates were processed through SDS–PAGE and transferred to polyvinylidene difluoride membranes. The membranes were blocked, incubated overnight with anti-HA (Merck Millipore #05-904) and anti-β-actin (Santa Cruz #sc-47778) primary antibodies, then with horseradish peroxidase (HRP)-conjugated secondary antibodies (Cell Signaling Technology #7076). Signals were detected using the Amersham ECL select kit on a Bio-Rad ChemiDoc MP system.

## AAV vector packaging and purification

As described previously (*He et al., 2022*), AAV vectors were produced in 293T cells transfected with AAV-nEF-sACE2-Fc, pAdDeltaF6 (Addgene #112867), and pAAV2/8 (Addgene #112864) plasmids using PEI. AAV particles were harvested from supernatants with polyethylene glycol 8000 and from cell lysates via freeze–thaw cycles and benzonase digestion, then purified using Iodixanol density gradient ultracentrifugation. AAV particles were concentrated and measured using a qPCR AAV Titer Kit (Applied Biological Materials #G931).

## SARS-CoV-2 virus

Experiments with live SARS-CoV-2 were performed at the BSL-3 core facility (LKS Faculty of Medicine, HKU). The BetaCoV/Hong Kong/VM20001061/2020 virus, here regarded as the wild-type strain of SARS-CoV-2 (Wuhan-Hu-1), was isolated from the nasopharyngeal aspirate and throat swab of a confirmed patient with COVID-19 in Hong Kong (GISAID identifier EPI_ISL_412028). The SARS-CoV-2 variants were isolated from clinical specimens in Hong Kong. Stock viruses were prepared with Vero E6 cells cultured in infection medium (DMEM supplemented with 2% FBS and 1% PS).

## Median tissue culture infectious dose (TCID$_{50}$) assay

Vero E6 cells pre-seeded in 96-well plates were infected with serially diluted virus stocks or mouse lung homogenates in infection medium. After 72-hr incubation, cytopathic effects (CPEs) were observed under a microscope to calculate titers using the Reed–Muench method.

## PRNT assay

Casirivimab (#C100P) and hIgG1 isotype (clone 4F17, #PA007125) were purchased from Syd Labs, USA, and used as controls. Purified B5-D3 and control proteins were serially diluted and incubated with 50 PFU of SARS-CoV-2 (Wuhan-Hu-1, Delta, Omicron BA.5, BQ.1.22, and XBB.1.5) for 1 hr at room temperature, followed by addition to Vero E6 cells seeded in 6-well plates. After incubation, cells were overlaid with agarose, fixed with formalin, and stained with crystal violet. Plaque counts were used to calculate percentage neutralization and half-maximal inhibitory concentration (IC$_{50}$) values. The experiments were carried out in duplicates.

## Animal experiments

The K18-hACE2 mice (B6.Cg-Tg(K18-ACE2)2Prlmn/J; RRID:IMSR_JAX:034860) (*McCray et al., 2007*) were purchased from the Jackson Laboratory (Bar Harbor, ME, USA). Experiments on AAV or protein-only administration in mice were carried out in the Animal Holding Core of the School of Biomedical Sciences, CUHK. Experiments involving SARS-CoV-2 infection in K18-hACE2 mice were conducted within the confines of the Biosafety Level 3 (BSL-3) core facility located at the Li Ka Shing Faculty of Medicine, HKU. Experiments were conducted according to ethical practices to minimize animal distress. All animal procedures were ethically approved by The Chinese University of Hong Kong (CUHK)'s Animal Experimentation Ethics Committee (approval number: 20-226-MIS) and The University of Hong Kong (HKU)'s Committee on the Use of Live Animals in Teaching and Research (approval number: 5511-20).

## AAV-mediated sACE2-Fc overexpression in mice

Male K18-hACE2 mice (*McCray et al., 2007*) at the age of 2 months were injected intravenously with $1 \times 10^{11}$ GC of AAV-nEF-sACE2-Fc. Blood samples were collected periodically for analysis. Following euthanasia, tissues were harvested for further DNA and histological analysis. sACE2-Fc concentrations in sera and RAS metabolites were quantified using specific ELISA kits for Human IgG1, Mouse Renin 1 (Invitrogen #EMREN1), Angiotensin II (LifeSpan BioSciences #LS-F523), and Angiotensin 1–7 (LifeSpan BioSciences #LS-F40645).

## SARS-CoV-2 infection in mice

Female K18-hACE2 mice, aged 10–12 or 2–3 months, were intranasally inoculated with $1 \times 10^4$ plaque-forming unit (PFU) of SARS-CoV-2 Wuhan-Hu-1. Treatment with B5-D3 protein was administered intranasally at 2.5 mg/kg or intravenously at 15 mg/kg, at various time points relative to the viral challenge (6 hr before, 24 hr before, or 24 hr after). Vehicle control groups received PBS 6 hr before

viral challenge. Survival and weight were monitored daily for 14 days. For the older mice, lung samples were collected from one mouse from each group at 4 days post-infection (dpi) for analysis; younger mice had plasma collected for neutralizing antibody analysis at 14 dpi. Another batch of young mice also received B5-D3(-LALA) protein pre-inoculation, with lungs analyzed post-inoculation for RNA, viral load, and histopathology. A control group of non-infected mice was used to assess baseline effects of B5-D3 on lung tissue.

## Neutralization assay for antibody titration

Vero E6 cells were pre-seeded on 96-well plates 24 hr before infection. On the day of infection, the growth medium of the cells was changed to infection medium. The plasma samples were serially twofold diluted with infection medium from a starting dilution of 1:10. The plasma was then pre-incubated with 100 $TCID_{50}$ of SARS-CoV-2 for 1 hr at room temperature before being inoculated to the seeded Vero E6 cells in quadruplicates. At 72 hr after inoculation, CPEs of the cells were observed with optical microscopy. Neutralizing antibody titers against SARS-CoV-2 were expressed as the reciprocal of the highest dilution of plasma showing no CPEs in all 4 wells. Uninfected cell monolayers were used as toxicity control.

## Histology

Mouse tissues were fixed in 10% formalin, embedded in paraffin, and sectioned at 5 µm. Sections of different organs were deparaffinized and underwent hematoxylin and eosin (H&E) staining. For IHC staining, lung sections underwent antigen retrieval, endogenous peroxidase blocking, and were incubated with primary antibodies against the SARS-CoV/SARS-CoV-2 nucleocapsid protein (Sino Biological #40143-T62) overnight. After washing, sections were stained with the anti-rabbit VECTASTAIN Elite ABC-HRP Kit (Vector Laboratories), developed with 3,3'-Diaminobenzidine (Sigma #D4293), and counterstained with Mayer's hematoxylin. Stained sections were scanned using Zeiss Axioscan 7 and analyzed using ZEN (blue edition). For measurement of alveolar septal thickness, 10 fields from each lung H&E section and 10 septa from each field were chosen for measurement using ImageJ.

## Quantitative PCR

Quantitative PCR (qPCR) was used to analyze liver genomic DNA and lung RNA from mice. DNA was extracted and qPCR was performed with the TB Green Premix Ex Taq II kit (Takara), normalized to mouse *Gapdh* using the $2^{-\Delta Ct}$ method. RNA was extracted using TRIzol, reverse-transcribed (Applied Biosystems #4368813), subjected to qPCR using the same kit, and normalized to mouse *Gapdh* levels. Specific primers are listed in *Supplementary file 1B*.

## RNA-Seq and data analysis

Total RNA was extracted from mouse lung tissues or THP-1-derived M0 macrophages using TRIzol and processed into transcriptome libraries with the TruSeq RNA Library Prep Kit (Illumina). Sequencing was performed on the NovaSeq 6000 or NovaSeq X Plus sequencers (Illumina) using a 150-base pair paired-end configuration. Sequencing data were processed with fastp for quality control (*Chen et al., 2018*), then aligned to both the mouse (Ensembl GRCm39) and SARS-CoV-2 (NCBI NC_045512v2) genomes using STAR (*Maulding et al., 2022*) or to the human genome (Ensembl GRCh38) using HISAT2 (*Kim et al., 2019*). Pearson correlation and groupwise comparisons were conducted in R: gene expression was quantified and analyzed for differential expression using DESeq2 (*Love et al., 2014*); up/downregulated gene enrichment and Gene Set Enrichment Analysis (GSEA) (*Subramanian et al., 2005*) was performed using the clusterProfiler package (*Wu et al., 2021*).

## Tracking B5-D3 bio-distribution in mice

B5-D3 were conjugated with Alexa Fluor 750 dye (B5-D3-AF750) as described (*Ku et al., 2021*). In brief, 2 mg/ml solution of B5-D3 protein in 0.15 M $NaHCO_3$ was reacted with Alexa Fluor 750 succinimidyl ester (Thermo Fisher Scientific) at room temperature for 1 hr. Unreacted dye was removed by dialysis in PBS. All procedures were performed under dimmed light. Female K18-hACE2 mice, aged 2–3 months, were administered intranasally with B5-D3-AF750 (2.5 mg/kg). The mice were imaged at predetermined time points after administration (fluorescence ex = 745 nm, em = 800 nm, auto-exposure setting) using an IVIS Spectrum CT Imager (Perkin Elmer). At the time of euthanasia, 50 µl of

urine and blood, the brain, nasal cavity, trachea, lung, heart, liver, spleen, kidney, and urinary bladder samples were excised and imaged. Regions of interest were drawn, and average radiance (p/s/cm²/sr) was measured. All images were processed using Living Image software (Perkin Elmer), and the same fluorescence threshold was applied for group comparison.

## Flow cytometry analysis and confocal microscopic imaging of BALF cells

Mice were sacrificed via anesthetic overdose. Bronchoalveolar lavage was performed by intratracheally rinsing the lungs with 1 ml of ice-cold Hanks' Balanced Salt Solution (Gibco) containing 100 μM ethylenediaminetetraacetic acid for four repeats. Bronchoalveolar lavage fluid (BALF) was then centrifuged and treated with ammonium-chloride-potassium red blood cell lysing buffer. For multicolor flow cytometry, cell pellets were washed with PBS and stained with the Fixable Viability Stain 440UV dye (BD #566332). Next, the cells were blocked with CD16/CD32 monoclonal antibody (Invitrogen #14-0161-85) and stained with antibodies targeting the following molecules: CD45 (BD #568336), Siglec-F (BD #564514), CD11b (BD #612800), CD11c (BD #751265), Ly6G (BD #563005), I-A/I-E major histocompatibility complex class II (MHC-II) (BD #750171), F4/80 (BD #570288), Ly6C (BD #755198), and CD3 (BD #555275). Stained BALF cells were analyzed using the BD FACSymphony A5.2 SORP Flow Cell Analyzer, and the results were analyzed using FlowJo v10.10.

For microscopic inspections, BALF cells were collected from mice, seeded in poly-D-lysine-coated chamber slides, and stained with antibodies targeting Siglec-F (BD #564514) and human IgG-Fc (Abcam #ab98596). Secondary antibody (Invitrogen #A-11006) was applied. Stained BALF cells were then counterstained with Hoechst and examined by confocal microscopy (Leica TCS SP8).

## Statistical Analysis

Assays including in vitro neutralization, PRNT, ADCC, and ADCP were conducted in technical duplicates. Results were analyzed in GraphPad Prism version 9 using nonlinear regression to calculate $IC_{50}$ or half-maximal effective concentration ($EC_{50}$) values. Transcriptomic analyses were performed using R, with details provided in figure captions. All other statistical analyses utilized GraphPad Prism version 9, with a significance threshold set at $p$ value ($p$) <0.05.

## Acknowledgements

This study was supported by Research Grants Council of Hong Kong grants 14115520, 14106024 (BF), C7145-20GF (LLP). The Centre for Regenerative Medicine and Health and The Centre for Immunology & Infection are supported by grants from the Health@InnoHK program, an initiative by the Innovation and Technology Commission of the Hong Kong SAR Government. Jingyi W, JL, BL, and JQ received postgraduate studentships from the Chinese University of Hong Kong. We thank the Chinese University of Hong Kong (CUHK) and the University of Hong Kong (HKU) research platforms for assistance in animal experimentation (the Laboratory Animal Service Center at CUHK and the Centre for Comparative Medicine Research at HKU) and histological analysis (Department of Pathology, HKU and Core Laboratory in the School of Biomedical Sciences, CUHK).

## Additional information

### Funding

| Funder | Grant reference number | Author |
| --- | --- | --- |
| Research Grants Council, University Grants Committee | 14115520 | Bo Feng |
| Research Grants Council, University Grants Committee | 14106024 | Bo Feng |

| Funder | Grant reference number | Author |
|---|---|---|
| Research Grants Council, University Grants Committee | C7145-20GF | Leo LM Poon |
| Innovation and Technology Commission | Health@InnoHK | Leo LM Poon<br>Bo Feng |
| Chinese University of Hong Kong | Postgraduate studentship | Jingyi Wang<br>Jiangchuan Li<br>Bin Luo<br>Jiale Qiu |

The funders had no role in study design, data collection, and interpretation, or the decision to submit the work for publication.

### Author contributions

Jingyi Wang, Conceptualization, Data curation, Software, Formal analysis, Validation, Investigation, Visualization, Methodology, Writing – original draft, Writing – review and editing; Jiangchuan Li, Data curation, Formal analysis, Validation, Investigation, Visualization, Methodology, Writing – review and editing; Alex WH Chin, Data curation, Formal analysis, Validation, Investigation, Visualization, Methodology, Writing – original draft; Bin Luo, Data curation, Validation, Methodology; Junkang Wei, Software, Formal analysis, Visualization; Jiale Qiu, Validation, Methodology; Jianwei Ren, Conceptualization, Supervision, Methodology; Yin Xia, Thomas Braun, Methodology, Writing – review and editing; Leo LM Poon, Supervision, Funding acquisition, Investigation, Methodology, Writing – review and editing; Bo Feng, Conceptualization, Resources, Supervision, Funding acquisition, Investigation, Methodology, Writing – original draft, Project administration, Writing – review and editing

### Author ORCIDs

Jingyi Wang https://orcid.org/0009-0009-4419-2618
Jiangchuan Li https://orcid.org/0009-0000-8932-3073
Alex WH Chin https://orcid.org/0000-0002-6556-9092
Yin Xia https://orcid.org/0000-0003-0315-7532
Thomas Braun https://orcid.org/0000-0002-6165-4804
Leo LM Poon https://orcid.org/0000-0002-9101-7953
Bo Feng https://orcid.org/0000-0002-4018-3257

### Ethics

All animal procedures were ethically approved by The Chinese University of Hong Kong (CUHK)'s Animal Experimentation Ethics Committee (approval number: 20-226-MIS) and The University of Hong Kong (HKU)'s Committee on the Use of Live Animals in Teaching and Research (approval number: 5511-20). Experiments were conducted according to ethical practices to minimize animal distress.

Reviewer #1 (Public review): https://doi.org/10.7554/eLife.108883.3.sa1
Reviewer #2 (Public review): https://doi.org/10.7554/eLife.108883.3.sa2
Reviewer #3 (Public review): https://doi.org/10.7554/eLife.108883.3.sa3
Author response https://doi.org/10.7554/eLife.108883.3.sa4

# Additional files

### Supplementary files

Supplementary file 1. Supplementary tables.

MDAR checklist

### Data availability

All data associated with this study are available in the main text or the supplementary materials. The RNA-seq data generated in this study have been deposited in the NCBI Sequence Read Archive database under accession code PRJNA1054508. Constructs of diverse sACE2-Fc mutants and SARS-CoV-2

spikes are available upon request after completion and approval of a material transfer agreement by contacting fengbo@cuhk.edu.hk.

The following dataset was generated:

| Author(s) | Year | Dataset title | Dataset URL | Database and Identifier |
|---|---|---|---|---|
| Wang J LiJ | 2026 | Study of engineered soluble ACE2 decoys in protecting against SARS-CoV-2 in mice | https://www.ncbi.nlm.nih.gov/bioproject/PRJNA1054508 | NCBI BioProject, PRJNA1054508 |

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

# Appendix 1

## Appendix 1—key resources table

| Reagent type (species) or resource | Designation | Source or reference | Identifiers | Additional information |
|---|---|---|---|---|
| Strain, strain background (*Mus musculus*) | K18-hACE2, B6.Cg-Tg(K18-ACE2)2Prlmn/J mice | The Jackson Laboratory | 034860, RRID:IMSR_JAX:034860 | |
| Cell line (*Homo sapiens*) | 293T | ATCC | CRL-3216 | |
| Cell line (*Cercopithecus aethiops*) | Vero E6 | ATCC | CRL-1586 | |
| Cell line (*Homo sapiens*) | Calu-3 | ATCC | HTB-55 | |
| Cell line (*Homo sapiens*) | THP-1 | ATCC | TIB-202 | |
| Cell line (*Homo sapiens*) | Expi293F | Gibco | A14527 (component of A14635) | For production of sACE2-Fc proteins |
| Cell line (*Homo sapiens*) | hACE2-293T | This paper | | 293T cells lentivirally transduced to overexpress full-length hACE2 for pseudovirus infection and neutralization assays; see in Materials and methods section under 'Lentivirus packaging and transduction' |
| Cell line (*Homo sapiens*) | hACE2-Calu-3 | This paper | | Calu-3 cells lentivirally transduced to overexpress full-length hACE2 as an in vitro model of lung epithelial cells; see in Materials and methods section under 'Lentivirus packaging and transduction' |
| Transfected construct (human) | HDM-SARS2-Spike-delta21 (plasmid) | Addgene | 155130 | Construct to express spike for lentiviral pseudotyping; backbone for sACE2(-Fc)-his candidate and spike variant subcloning |
| Transfected construct (human) | HDM-CMV-sACE2(-Fc)-his and mutants (plasmids) | This paper | | Constructs to transfect and express the sACE2(-Fc) candidate proteins; template for site-directed mutagenesis to generate mutant sACE2-Fc (A2, A3, B2–B6, D1–D5, or combinations); see in Materials and methods section under 'Plasmid construction' |
| Transfected construct (human) | HDM-CMV-sACE2-Fc-LALA-his (plasmid) | This paper | | Construct to transfect and express B5-D3-LALA protein; see in Materials and methods section under 'Plasmid construction' |

*Appendix 1 Continued on next page*

*Appendix 1 Continued*

| Reagent type (species) or resource | Designation | Source or reference | Identifiers | Additional information |
|---|---|---|---|---|
| Transfected construct (human) | AAV-nEF-sACE2-Fc double mutants (plasmids) | This paper | | AAV constructs to transfect and express sACE2-Fc double mutants; see in Materials and methods section under 'Plasmid construction' |
| Transfected construct (human) | psPAX2 (plasmid) | Addgene | 12260 | Lentiviral packaging plasmid |
| Transfected construct (human) | pMD2.G (plasmid) | Addgene | 12259 | Lentiviral packaging plasmid |
| Transfected construct (human) | pWPI-IRES-Puro-Ak-ACE2-TMPRSS2 (plasmid) | Addgene | 154987 | Lentiviral construct to transfect and express hACE2 and template of coding sequence of hACE2 |
| Transfected construct (human) | pCDH-EF1a-eFFly-eGFP (plasmid) | Addgene | 104834 | Lentiviral construct to transfect and express luciferase and GFP |
| Transfected construct (human) | pBOB-CAG-SARS-CoV-2-Spike-HA | Addgene | 141347 | Construct to transfect and express full-length spike with HA tag |
| Biological sample (lentiviral vectors) | Lenti-hACE2 | This paper | | Lentiviral vector to transduce and express full-length hACE2; see in Materials and methods section under 'Lentivirus packaging and transduction' |
| Biological sample (lentiviral vectors) | SARS-CoV-2 pseudovirus | This paper | | See in text for detailed mutations in spike; see in Materials and methods section under 'Pseudovirus packaging, titration, and infection' |
| Biological sample (AAV vectors) | AAV-sACE2-Fc | This paper | | Vectors for in vivo overexpression of sACE2-Fc double mutants. See in text for detailed mutations; see Materials and methods section under in 'AAV vector packaging and purification' |
| Biological sample (SARS-CoV-2) | Wuhan-Hu-1 | Prof. Leo Poon's lab | BetaCoV/Hong Kong/VM20001061/2020 | Isolated from a confirmed patient with COVID-19 in Hong Kong, GISAID identifier: EPI_ISL_412028 |
| Biological sample (SARS-CoV-2) | Delta virus | Prof. Leo Poon's lab | | Isolated from clinical specimens in Hong Kong |
| Biological sample (SARS-CoV-2) | Omicron BA.5 virus | Prof. Leo Poon's lab | | Isolated from clinical specimens in Hong Kong |
| Biological sample (SARS-CoV-2) | Omicron BQ.1.22 virus | Prof. Leo Poon's lab | | Isolated from clinical specimens in Hong Kong |

*Appendix 1 Continued on next page*

*Appendix 1 Continued*

| Reagent type (species) or resource | Designation | Source or reference | Identifiers | Additional information |
|---|---|---|---|---|
| Biological sample (SARS-CoV-2) | Omicron XBB.1.5 virus | Prof. Leo Poon's lab | | Isolated from clinical specimens in Hong Kong |
| Antibody | anti-ACE2 (Rabbit polyclonal) | Abcam | Cat# ab15348, RRID:AB_301861 | IF(1:500) |
| Antibody | anti-Rabbit secondary antibody, Alexa Fluor 594 (Chicken polyclonal) | Invitrogen | Cat# A-21442, RRID:AB_2535860 | IF(1:500) |
| Antibody | anti-human IgG-Fc, PE (Goat polyclonal) | Abcam | Cat# ab98596, RRID:AB_10673825 | IF(1:500) |
| Antibody | anti-HIV-1 p24 (Rabbit polyclonal) | Invitrogen | Cat# PA5-81773, RRID:AB_2788949 | IF(1:200) |
| Antibody | anti-LAMP1 (Mouse monoclonal) | Abcam | Cat# ab25630, RRID:AB_470708 | IF(1:100) |
| Antibody | anti-Mouse secondary antibody, Alexa Fluor 488 (Chicken polyclonal) | Invitrogen | Cat# A-21200, RRID:AB_2535786 | IF(1:500) |
| Antibody | anti-Rabbit secondary antibody, Alexa Fluor 647 (Donkey polyclonal) | Invitrogen | Cat# A-31573, RRID:AB_2536183 | IF(1:500) |
| Antibody | anti-HA (Mouse monoclonal) | Merck Millipore | Cat# 05–904, RRID:AB_417380 | WB(1:1000) |
| Antibody | anti-β-actin (Mouse monoclonal) | Santa Cruz | Cat# sc-47778, RRID:AB_626632 | WB(1:1000) |
| Antibody | anti-Mouse secondary antibody, HRP (Horse polyclonal) | Cell Signaling Technology | Cat# 7076, RRID:AB_330924 | WB(1:5000) |
| Antibody | anti-SARS-CoV/SARS-CoV-2 nucleocapsid protein (Rabbit polyclonal) | Sino Biological | Cat# 40143-T62, RRID:AB_2892769 | IHC(1:1000) |
| Antibody | anti-CD16/CD32 (Mouse monoclonal) | Invitrogen | Cat# 14-0161-85, RRID:AB_467134 | Flow cytometry(1 μl per test) |
| Antibody | anti-CD45 (Rat monoclonal) | BD | Cat# 568336, RRID:AB_3684191 | Flow cytometry(2 μl per test) |
| Antibody | anti-Siglec-F (Rat monoclonal) | BD | Cat# 564514, RRID:AB_2738833 | Flow cytometry(0.5 μl per test), IF(1:50) |
| Antibody | anti-CD11b (Rat monoclonal) | BD | Cat# 612800, RRID:AB_2738811 | Flow cytometry(1 μl per test) |
| Antibody | anti-CD11c (Hamster monoclonal) | BD | Cat# 751265, RRID:AB_2875281 | Flow cytometry(1 μl per test) |
| Antibody | anti-Ly6G (Rat monoclonal) | BD | Cat# 563005, RRID:AB_2737946 | Flow cytometry(1 μl per test) |
| Antibody | anti-MHC-II (Rat monoclonal) | BD | Cat# 750171, RRID:AB_2874376 | Flow cytometry(0.5 μl per test) |
| Antibody | anti-F4/80 (Rat monoclonal) | BD | Cat# 570288, RRID:AB_3678614 | Flow cytometry(0.5 μl per test) |
| Antibody | anti-Ly6C (Rat monoclonal) | BD | Cat# 755198, RRID:AB_3099650 | Flow cytometry(0.5 μl per test) |
| Antibody | anti-CD3 (Rat monoclonal) | BD | Cat# 555275, RRID:AB_395699 | Flow cytometry(2 μl per test) |
| Antibody | anti-Rat secondary antibody, Alexa Fluor 488 (Goat polyclonal) | Invitrogen | Cat# A-11006, RRID:AB_2534074 | IF(1:500) |
| Recombinant DNA reagent | pGEM-T Easy Vector | Promega | A1360 | Backbone for hACE2 subcloning and site-directed mutagenesis |

*Appendix 1 Continued on next page*

*Appendix 1 Continued*

| Reagent type (species) or resource | Designation | Source or reference | Identifiers | Additional information |
|---|---|---|---|---|
| Recombinant DNA reagent | pcDNA3-SARS-CoV-2-S-RBD-Fc (plasmid) | Addgene | 141183 | Template of coding sequence of human IgG1 hinge-Fc regions |
| Recombinant DNA reagent | pAAV-nEFCas9 (plasmid) | Addgene | 87115 | Backbone for sACE2-Fc double mutant subcloning |
| Peptide, recombinant protein | sACE2 protein | This paper | | sACE2 (aa 1–740) without Fc; see in Materials and methods section under 'Protein production and purification' |
| Peptide, recombinant protein | sACE2-Fc proteins | This paper | | See in text for detailed mutations; see in Materials and methods section under 'Protein production and purification' |
| Peptide, recombinant protein | B5-D3-LALA protein | This paper | | sACE2-Fc B5-D3 variant with LALA mutations; see in Materials and methods section under 'Protein production and purification' |
| Peptide, recombinant protein | Casirivimab | Syd Labs | C100P | |
| Peptide, recombinant protein | hIgG1 isotype, clone 4F17 | Syd Labs | PA007125 | |
| Commercial assay or kit | Lenti-X qRT-PCR Titration Kit | Takara | 631235 | |
| Commercial assay or kit | Luciferase Assay System | Promega | E1501 | |
| Commercial assay or kit | ExpiFectamine 293 Transfection Kit | Gibco | A14524 | |
| Commercial assay or kit | Human ACE2 ELISA Kit | Abcam | ab235649 | |
| Commercial assay or kit | Human IgG1 ELISA Kit | Invitrogen | BMS2092 | |
| Commercial assay or kit | Angiotensin II Converting Enzyme (ACE2) Activity Assay Kit (Fluorometric) | Abcam | ab273297 | |
| Commercial assay or kit | Ni-NTA Agarose | QIAGEN | 30210 | |
| Commercial assay or kit | Bio-Rad Protein Assay | Bio-Rad | 5000001 | Bradford protein assay |
| Commercial assay or kit | Jurkat-Lucia NFAT-CD16 cells | InvivoGen | jktl-nfat-cd16 | ADCC reporter assay |
| Commercial assay or kit | Jurkat-Lucia NFAT-CD32 cells | InvivoGen | jktl-nfat-cd32 | ADCP reporter assay |
| Commercial assay or kit | QUANTI-Luc | InvivoGen | rep-qlc4lg1 | Detection of luminescence in ADCC and ADCP reporter assays |
| Commercial assay or kit | TRIzol Reagent | Invitrogen | 15596026 | |
| Commercial assay or kit | Amersham ECL Select Western Blotting Detection Reagent | Cytiva | RPN2235 | |
| Commercial assay or kit | qPCR AAV Titer Kit | Applied Biological Materials | G931 | |
| Commercial assay or kit | Mouse Renin 1 (REN1) ELISA Kit | Invitrogen | EMREN1 | |
| Commercial assay or kit | Mouse/Human/Rat Angiotensin II ELISA Kit | LifeSpan BioSciences | LS-F523 | |
| Commercial assay or kit | Mouse Angiotensin 1–7 ELISA Kit | LifeSpan BioSciences | LS-F40645 | |

*Appendix 1 Continued on next page*

*Appendix 1 Continued*

| Reagent type (species) or resource | Designation | Source or reference | Identifiers | Additional information |
|---|---|---|---|---|
| Commercial assay or kit | VECTASTAIN Elite ABC-HRP Kit, Peroxidase (Rabbit IgG) | Vector Laboratories | PK-6101 | |
| Commercial assay or kit | TB Green Premix Ex Taq II kit | Takara | RR82WR | |
| Commercial assay or kit | High-Capacity cDNA Reverse Transcription Kit | Applied Biosystems | 4368813 | |
| Commercial assay or kit | TruSeq RNA Sample Prep Kit | Illumina | FC-122–1001 | |
| Chemical compound, drug | 3,3'-Diaminobenzidine | Sigma | D4293 | |
| Chemical compound, drug | Alexa Fluor 750 NHS Ester (Succinimidyl Ester) | Invitrogen | A37575 | For fluorescent labeling of B5-D3 protein |
| Other | Normal Goat Serum | Thermo Scientific | 50062Z | 10% |
| Other | Hoechst 33342 stain | Thermo Scientific | 62249 | (1 µg/ml) |
| Other | Fixable Viability Stain 440UV dye | BD | 566332 | Flow cytometry(1:500) |

