## [Editor Report · eLife Assessment]

This manuscript presents a **valuable** antiviral approach using an engineered ACE2-Fc fusion protein that demonstrates broad-spectrum neutralization capacity against SARS-CoV-2 variants and achieves significant prophylactic protection in animal models through a novel Fc-mediated phagocytosis mechanism. The study provides **convincing** evidence for protective efficacy through rigorous in vivo validation in mice, mechanistic characterization via transcriptomic analysis and biodistribution studies, and demonstration of antibody-dependent cellular phagocytosis as the primary clearance mechanism mediated by the decoy. The work will be of interest to researchers working in vaccine development and associated immune responses.

---

## [Referee Report · Reviewer #1 (Public review)]

Summary:

This manuscript by Wang et al. describes the development of an optimized soluble ACE2-Fc fusion protein, B5-D3, for intranasal prophylaxis against SARS-CoV-2. As shown, B5-D3 conferred protection not only by acting as a neutralizing decoy, but also by redirecting virus-decoy complexes to phagocytic cells for lysosomal degradation. The authors showed complete in vivo protection in K18-hACE2 mice and investigated the underlying mechanism by a combination of Fc-mutant controls, transcriptomics, biodistribution studies, and in vitro assays.

Strengths:

The major strength of this work is the identification of a novel antiviral approach with broad-spectrum and beyond simple neutralization. Mutant ACE2 enables broad and potent binding activity with the S proteins of SARS-CoV-2 variants, while the fused Fc part mediates phagocytosis to clear the viral particles. The conceptual advance of this ACE2-Fc combination is convincingly validated by in vivo protection data and by the completely abrogated protection of Fc LALA mutant.

Additionally:

The authors include a discussion (in Discussion part) about a previously reported ACE2 decamer (DOI: 10.1080/22221751.2023.2275598) and compared with the ACE2-Fc fusion protein developed in this study. The authors also tested the off-target activity and showed no evidence of toxicity in vivo.

---

## [Referee Report · Reviewer #2 (Public review)]

Summary:

Wang et al. engineered an ACE2 mutant by introducing two mutations (T92Q and H374N), and fused this ACE2 mutant to human IgG1-Fc (B5-D3). Experimental results suggest that B5-D3 exhibits broad-spectrum neutralization capacity and confers effective protection upon intranasal administration in SARS-CoV-2-infected K18-hACE2 mice. Transcriptomic analysis suggests that B5-D3 induces early immune activation in lung tissues of infected mice. Fluorescence-based bio-distribution assay further indicates rapid accumulation of B5-D3 in the respiratory tract, particularly in airway macrophages. Further investigation shows that B5-D3 promotes viral phagocytic clearance by macrophages via an Fc-mediated effector function, namely antibody-dependent cellular phagocytosis (ADCP), while simultaneously blocking ACE2-mediated viral infection in epithelial cells. These results provide some insights into improving decoy treatments against SARS-CoV-2 and other potential respiratory viruses.

Strengths:

The protective effect of this ACE2-Fc fusion protein against SARS-CoV-2 infection has been evaluated in a reasonable way.

Weaknesses:

(1) Some of the mice experiments suffer from insufficient sample numbers, which affect the statistical power and reliability of the results. The author acknowledged this weakness, noting that the supply of aged mice was limited, while arguing that, although the sample size is small, the data from these mice are consistent.

(2) Compared to 6 hours, intranasal administration of B5-D3 at 24 hours before viral infection results in reduced protective efficacy. However, only survival and body weight data are provided, with no supporting evidence from virological assays such as viral titer measurement. The author acknowledged that such data would be more comprehensive and attributed the limitation to constraints in animal services.

(3) The efficacy of the B5-D3-LALA group was not as good as that of the B5-D3 group. The author suggested that there might be a certain degree of viral variation, and viral infection in the lungs may be uneven in the B5-D3-LALA group.

---

## [Referee Report · Reviewer #3 (Public review)]

Strengths:

The core strength of this study lies in its innovative demonstration that an engineered sACE2-Fc fusion redirects virus-decoy complexes to Fc-mediated phagocytosis and lysosomal clearance in macrophages, revealing a distinct antiviral mechanism beyond traditional neutralization. Its complete prophylactic protection in animal models and precise targeting of airway phagocytes establish a novel therapeutic paradigm against SARS-CoV-2 variants and future respiratory viruses.

Weaknesses:

The study attributes the complete antiviral protection to Fc-mediated phagocytic clearance, a central claim that requires more rigorous experimental validation. The observation that abrogating Fc functions compromises protection could be confounded by potential alterations in the protein's stability, half-life, or overall structure. To firmly establish this mechanism, it is crucial to include a control molecule with a mutated Fc region that lacks FcγR binding while preserving the Fc structure itself. Without this critical control, the conclusion that phagocytic clearance is the primary mechanism remains inadequately supported. The strategy of deliberately targeting virus-decoy complexes to phagocytes via Fc receptors inherently raises the question of Antibody-Dependent Enhancement (ADE) of disease. While the authors demonstrate a lack of productive infection in macrophages, this only addresses one facet of ADE. The risk of Fc-mediated exacerbation of inflammation (ADE) remains a critical concern. The manuscript would be significantly strengthened by a direct discussion of this risk and by including data, such as cytokine profiling from treated macrophages, to more comprehensively address the safety profile of this approach. The exclusive use of the K18-hACE2 mouse model, which exhibits severe disease, limits the generalizability of the findings. The "complete protection" observed may not translate to models with more robust and naturalistic immune responses or to human physiology. Furthermore, the lack of data against circulating SARS-CoV-2 variants of concern. The concept of sACE2-Fc fusion proteins as decoy receptors is not novel, and numerous similar constructs have been previously reported. The manuscript would benefit from a clearer demonstration of how the optimized B5-D3 mutant represents a significant advance over existing sACE2-Fc designs. A direct comparative analysis with previously published benchmarks, particularly in terms of neutralizing potency, Fc effector function strength, and in vivo efficacy, is necessary to establish the incremental value and novelty of this specific agent.

Comments on revised version:

The author has successfully addressed the raised issue.

---

## [Author Response]

The following is the authors’ response to the original reviews.

**Public Reviews:**

**Reviewer #1 (Public review):**
Summary:This manuscript by Wang et al. describes the development of an optimized soluble ACE2-Fc fusion protein, B5-D3, for intranasal prophylaxis against SARS-CoV-2. As shown, B5-D3 conferred protection not only by acting as a neutralizing decoy, but also by redirecting virus-decoy complexes to phagocytic cells for lysosomal degradation. The authors showed complete in vivo protection in K18-hACE2 mice and investigated the underlying mechanism by a combination of Fc-mutant controls, transcriptomics, biodistribution studies, and in vitro assays.Strengths:The major strength of this work is the identification of a novel antiviral approach with broad-spectrum and beyond simple neutralization. Mutant ACE2 enables broad and potent binding activity with the S proteins of SARS-CoV-2 variants, while the fused Fc part mediates phagocytosis to clear the viral particles. The conceptual advance of this ACE2-Fc combination is convincingly validated by in vivo protection data and by the completely abrogated protection of Fc LALA mutant.

We thank the reviewer for his recognition and positive comments on our study.

Weaknesses:Some aspects could be further modified.(1) A previously reported ACE2 decamer (DOI: 10.1080/22221751.2023.2275598) needs to be mentioned and compared in the Discussion part.

We thank the reviewer for pointing out this weakness.

Indeed, previous studies reported that the ACE2-IgM decamer, taking advantage of the decameric structure of IgM, exhibited higher avidity to spikes and greater potency for viral neutralization [1-3]. In particular, the study by Guo et al. has demonstrated a broad-spectrum neutralization ability of the ACE2-IgM decamer against multiple SARS-CoV-2 variants and reported the efficacy of intranasal prophylaxis in preventing lethal SARS-CoV-2 challenge in K18-hACE2 mice.

We agree with the reviewer that it is promising that our B5-D3 design would benefit from switching to the IgM isotype. However, the distinct biological features imposed by IgM Fc, including short serum half-life and restricted tissue penetration [4], may complicate the study design and diverge our focus.

In our current study, we would focus on the IgG1 Fc-based decoy design, while inactivating the enzyme activity of ACE2 to avoid disturbing the renin angiotensin system. This design allowed us to compare diverse administration routes and regimens and to gain useful insights into the potential of sACE2-Fc decoy in combating SARS-CoV-2 in vivo.

We appreciated the reviewer‘s insightful suggestion. In the revised manuscript, we have included additional discussion regarding ACE2-IgM decamer, addressing the relevant concern on page 17 lines 409–414.

(2) Limitations of this study, such as off-target binding and potential immunogenicity, should also be discussed.

We thank the reviewer for his insightful comments and agree that off-target activity is a major concern for designing the ACE2 decoy.

(1) In our study, the representative sACE2-Fc decoy candidate B5-D3 contains H374N mutation (D3) that is designed to inactivate ACE2 enzyme activity by causing dyscoordination of Zn2+. Our in vitro enzymatic activity assay has demonstrated that the H374N mutation (D3), as well as other three single mutations D1, D4 and D5, in either WT sACE2-Fc or B5 mutant, could effectively abolish the hACE2 enzyme activity (Supplementary Fig. 2e, h).

(2) To further address the concern on off-target activity, we performed AAV-based overexpression experiments in K18-hACE2 mice and examined serum levels of RAS hormones, using ELISA methods that specifically detect serum renin, Angiotensin II (Ang II), and Ang (1-7). While our data from WT sACE2-Fc overexpression revealed significantly elevated serum renin and Ang II, indicating a disruption of the RAS (Supplementary Fig. 4d, e); the results from examined double mutants, including B5-D3, showed negligible change in any of these metabolite levels, demonstrating no off-target effect and minimal disturbance to the RAS activity in K18-hACE2 mice (Supplementary Fig. 4d–f).

(3) Moreover, in this experiment, after the prolonged overexpression of all these molecules in K18hACE2 mice, histological examination of multiple organs showed no evidence of immune cell infiltration and tissue damage and no difference was observed between the mice receiving WT sACE2-Fc or B5-D3(Supplementary Fig. 4g).

In the revised manuscript, we have included the results from the AAV-delivered in vivo overexpression of WT sACE2-Fc and three most promising double mutants (B5-D3, B5-D4 and B5-D5) on page 5 lines 118–122 and on page 6 lines 123–135 in the main text. The relevant data were presented in the new Supplementary Fig. 4.

**Reviewer #2 (Public review):**
Summary:Wang et al. engineered an optimized ACE2 mutant by introducing two mutations (T92Q and H374N) and fused this ACE2 mutant to human IgG1-Fc (B5-D3). Experimental results suggest that B5-D3 exhibits broad-spectrum neutralization capacity and confers effective protection upon intranasal administration in SARS-CoV-2-infected K18-hACE2 mice. Transcriptomic analysis suggests that B5D3 induces early immune activation in lung tissues of infected mice. Fluorescence-based biodistribution assay further indicates rapid accumulation of B5-D3 in the respiratory tract, particularly in airway macrophages. Further investigation shows that B5-D3 promotes viral phagocytic clearance by macrophages via an Fc-mediated effector function, namely antibody-dependent cellular phagocytosis (ADCP), while simultaneously blocking ACE2-mediated viral infection in epithelial cells. These results provide insights into improving decoy treatments against SARS-CoV-2 and other potential respiratory viruses.Strengths:The protective effect of this ACE2-Fc fusion protein against SARS-CoV-2 infection has been evaluated in a quite comprehensive way.

We thank the reviewer for his recognition and positive comments on our study.

Weaknesses:(1) The paper lacks an explanation regarding the reason for the combination of mutations listed in Supplementary Figure 2b. For example, for the mutations that enhance spike protein binding, B2-B6 does not fully align with the mutations listed in Table S1 of Reference 4, yet no specific criteria are provided.

We thank the reviewer for pointing out this negligence.

We constructed the B2-B6 mutants based on the study by Chan et al. [5] (Reference 4 in the previous version), mainly referencing to their Fig. 1A rather than to their Table S1. In Chan’s study, each of the proposed mutations were discovered as single mutations in monomeric sACE2 molecules based on the enrichment in target cell-binding. T92 was a notable hot spot for enriched mutations in their Fig. 1A.

Since monomeric and dimeric forms of sACE2 showed dramatically different kinetics for ACE2-RBD interaction, we selected five proposed mutations and further examined their affinity and activity in dimeric sACE2-Fc in our study. We chose not only the combinations of mutations, such as B3, B4, and B6 proposed in their Table S1, but also explored less-complicated mutation(s) like B2 (T27Y/L79T) and B5 (T92Q) in their Fig. 1A, which were *in silico* predicted to enhance ACE2-RBD binding but not tested in sACE2-Fc in Chan’s study.

Interestingly, although our results confirmed enhanced viral neutralization by all these mutations, the activity increase compared to WT ACE2-Fc was rather limited. Hence, we chose not to explore other mutations but to focus on B2–B6 to construct an enhanced ACE2-Fc decoy as a representative, to investigate the potential of ACE2-Fc decoys in combating SARS-CoV-2 infections.

In the revised manuscript, we have further amended the writing on page 4 lines 84–87 to enhance the readability. Whereas for conciseness of the manuscript, we did not describe in too much detail how we selected the mutations to be tested.

Second, for the mutations that abolished enzymatic activity, while D1 and D2, D3, D4, and D5 are cited from References 12, 11, and 33, respectively, the reason for combining D3 and D4 into A2, and D1 and D2 into A3 remains unexplained. It is also unclear whether some of these other possible combinations have been tested. Furthermore, for the B5-derived mutations, only double-mutant combinations with D1-D5 are tested, with no attempt made to evaluate triple mutations involving A2 or A3.

We thank the reviewer for pointing out this negligence.

A2 and A3 mutations were originally proposed as double mutations [6,7]. A2 (H374N/H378N) was first reported by Guy et al. [6] (Reference 11 in the previous version), while A3 (R273G/T445G) was originally proposed in Payandeh et al.’s study [7] (Reference 33 in the previous version).

In this study, we further split the two mutations in A2 and A3, to generate the single enzymedeactivating mutations, D1 and D2 from A3, and D3 and D4 from A2. Among these single mutations, D2 failed to inactivate ACE2 enzymatic activity (Supplementary Fig. 2e), and it was excluded in subsequent analyses.

D5 (H345L) was a single mutation directly adopted from the report by Glasgow et al. [8] (Reference 12 in the previous version).

After combining the B5 with the enzyme-deactivating mutations (A2, A3, D1, D3, D4, D5), our neuralization assay results showed that, the simpler compound mutants with only two mutations, like B5-D1, B5-D3, B5-D4 and B5-D5, exhibited stronger neutralization capacity than B5-A2 and B5-A3 with triple mutations. Moreover, since fewer mutations were more favorable to reduce risks in causing protein structure alteration and evoking host immunity, we then focused on the sACE2-Fc double mutants B5-D3, B5-D4 and B5-D5 in the subsequent neutralization and overexpression assays (Supplementary Fig. 3 and 4), and examined B5-D3 as a representative candidate in the in vivo infection tests and follow-up analysis (Figure 2–6, and Supplementary Figures 5–18).

We agree that the lack of explanation for splitting A2 and A3 into D1 to D4 single mutations made the rationale unclear. In the revised manuscript, we have included our previous test results on B5-A2 and B5-A3, cited Lei et al.’s study using A2 in ACE2 decoy [9], and explained the rationale for splitting A2 and A3 into D1 to D4 mutations. Relevant revision was made on page 4 lines 94–97 in the main text, while the design and data for B5-A2 and B5-A3 were included in the revised Figure 1b and Supplementary Figure 2b, f–h.

(2) Figures 1b, 1d, and 1e lack statistical analyses, making it difficult to determine whether B5 and D3 exhibit significant advantages. For Wuhan-Hu-1 strain, B2 and B5 are similar, and for D614G strain, B2, B3, B4, B5, and B6 display comparable results. However, only the glycosylation-related single mutant B5 is chosen for further combinatorial constructs. Moreover, for VOC/VOI strains, B5 is superior to B5-D3; for the Alpha strain, B5-D4 and B5-D5 are superior to B5-D3; and for the Delta and Lambda strains, B5-D5 is superior to B5-D3. These observations further highlight the need for a clearer explanation of the selection strategy.

We agree with the reviewer’s insightful observations.

Indeed, although our results confirmed enhanced viral neutralization by these reported mutations, the activity increases compared to WT ACE2-Fc were generally limited. Importantly, these observations were largely consistent with other reports (including the study by Chan et al. [5]), suggesting limited potential of mutagenesis in enhancing the ACE2-RBD/Spike interaction. Therefore, we chose to selectively examine B2-B6 to construct an enhanced ACE2-Fc decoy with reasonable performance, as a representative candidate to study the application potential of ACE2-Fc decoy.

The IC_50_ values in Figures 1b, 1d, and 1e were calculated from neutralization curves, measuring infection reduction at multiple concentrations in duplicates, which therefore were presented with statistical support. Based on the multiple neutralization assays, B5-D3 consistently showed a high performance among other top-performers (Figure 1, Supplementary Fig. 2f,g, and Supplementary Fig. 3).

We agree that B2 and B5 performed comparably well in neutralization assays, but B2 contains two mutations (T27Y/T92Q) while B5 carries a single mutation (T92Q). Hence, we decided to focus on B5 due to its lowest mutational burden and least potential risk.

We agree that for VOC/VOI strains, B5 was superior to B5-D3 in pseudovirus-neutralization assays. However, B3-D3 was enzymatically inactive, which is essential for generating safe ACE2 decoy and, therefore, justifies our usage of B5-D3 over B5.

We agree with the reviewer that, altogether, the B5-D3 did not show significant advantages than other top performers like B5-D4 and B5-D5. Here, B5-D3 was selected as a representative, which performed equally well rather than being the most outstanding candidate, for subsequent examination of efficacy, safety, and mechanistic insights.

We thank the reviewer for his valuable feedback. In the revised manuscript, we have further amended our description of B5-D3, as a “representative” candidate, to improve the readability. Relevant changes can be found on page 4 line 84, page 5 line 109, page 14 line 333 and page 15 line 360.

(3) Figure 1e does not specify the construct form of the control hIgG1, namely whether it is an hIgG1 Fc fragment or a full-length hIgG1 protein. If the full-length form is used, the design of its Fab region should be clarified to ensure the accuracy and comparability of the experimental control.

We thank the reviewer for pointing out this negligence.

In this study, we used the in vivo grade recombinant human IgG1 isotype control antibody in its full length (Syd labs, #PA007125) as the negative control. It is the 4F17 clone, which is widely used and showed low or no specific binding to any human samples [10] (Human IgG1 Isotype Control Antibody | Recombinant, in vivo Grade - Syd Labs). We have added the relevant information in the MATERIALS AND METHODS on page 23 lines 548–549.

(4) In Figure 2a, all three PBS control mice died, whereas in Figure 2f, three out of five PBS control mice died, with the remaining showing gradual weight recovery. This discrepancy may reflect individual immune variations within the control groups, and it is necessary to clarify whether potential autoimmune factors could have affected the comparability of the results. Also, the mouse experiments suffer from insufficient sample sizes, which affects the statistical power and reliability of the results. In Figure 2a, each group contains only 4 replicates, one of which was used for lung tissue sampling. As a result, body weight monitoring data is derived from only 3 mice per group (the figure legend indicating n=4 should be corrected to n=3). Such a small sample size limits the robustness of the conclusions. Similarly, in Figure 2f, although each group has 5 replicates, body weight data are presented for only 4 mice, with no explanation provided for the exclusion of the fifth mouse. Furthermore, the lung tissue experiments in Figure 3a include only 3 replicates, which is also inadequate.

We thank the reviewer for his valuable feedback.

Figure 2a was the first in vivo infection experiment of this study, and we performed the test in aged female K18-hACE2 mice at 10–12 months old. Whereas for the subsequent experiments in Figure 2f and Figure 3, we changed to young female K18-hACE2 mice at 2–3 months old, because the limited supply of old mice. While in Figure 2a, four aged mice (not three) in the PBS control group all died within 7 dpi, results of Figure 2f and Figure 3 consistently showed heterogeneous responses among young mice in the PBS control groups. Since increased susceptibility to SARS-CoV-2 infection has been broadly observed among aged human populations and it was also supported by mouse study [11], here we would attribute the observed discrepancy to the age difference between the two cohorts in Figure 2a and 2f. In the revised manuscript, we have further elucidated this observation in results (on page 7 lines 163–167) and included a new reference for better clarification (page 7 line 167).

Furthermore, because the PBS control mice in both Figure 2a and 2f died within 7 dpi, which was too soon for autoimmune factors to take place. Moreover, we have performed AAV-based prolonged overexpression experiments in K18-hACE2 mice (new Supplementary Fig. 4), which showed no tissue damage in either WT sACE2-Fc or B5-D3 treated mice, suggesting low immunogenicity. Collectively, the autoimmune factors are unlikely the reason leading to the different survival between PBS controls in Figure 2a and 2f.

We thank the reviewer for pointing out the weakness regarding small sample sizes in our study.

(1) In Figure 2a–c, the experiment was performed in an aged cohort at 10–12 months old, starting with 5 mice in each virus-inoculated group and 4 mice in the mock control group. At 4 dpi, we sacrificed one mouse from each group for tissue analysis. Therefore, in the survival analysis, there were 4 mice in each virus-inoculated group and 3 mice in the mock control group, whose survival and body weight changes were presented in Figure 2b, c.

Despite the relatively small sample sizes in Figure 2b, c, all 4 PBS control mice died, while all 4 mice in 6-hour B5-D3 IN prophylaxis group survived, demonstrating 100% survival and no sign of body weight loss. The survival and body weight data were highly consistent, strongly supporting that B5-D3 intranasal prophylaxis could protect the mice from lethal SARS-CoV-2 infection.

To enhance clarity, in the revised manuscript, we have added the sample size information in chart legends in Figure 2a–c.

(2) In Figure 2f–h, the experiment was performed in a young cohort at 2–3 months old and the body weight and survival data were presented for 5 mice in each group (not for 4 mice). Notably, although 2 out of 5 young mice in the PBS control group eventually survived from the viral infection, they had suffered significant weight loss during 4–7 dpi, similarly to the died. Whereas all 5 mice in the – 6hr B5-D3 IN prophylaxis group showed no sign of weight loss. Hence, these data were highly consistent with Figure 2b, c, supporting the efficiency of B5-D3 IN prophylaxis in protection against SARS-CoV-2 infection.

We noticed that some data points in Figure 2g, h were very close to each other, making it difficult to distinguish the data line for individual mice. To enhance clarity, in the revised manuscript, we have added sample-size information in chart legends in Figure 2g and 2h.

(3) In Figure 3a, we aimed to examine the lung tissues at early time points. For each treatment, we have 3 mice sacrificed at a single selected time point. Hence, total 9 mice were examined in the PBS control group and B5-D3 IN group, yielding results at 1 dpi, 2 dpi and 4 dpi that consistently supported each other. Moreover, the viral titers, S, and N protein expression analysis all showed significant difference among different groups. Therefore, our experiments have enough discrepancy between different treatment groups to draw the conclusion.

(5) Compared to 6 hours, intranasal administration of B5-D3 at 24 hours before viral infection results in reduced protective efficacy. However, only survival and body weight data are provided, with no supporting evidence from virological assays such as viral titer measurement. Therefore, the long-term effectiveness lacks sufficient experimental validation.

In Figure 2f–h, we aimed to compare the efficacies of IN administration of B5-D3 at different timepoints, mainly focusing on the body weight change and survival data along the infection and recovery time. As indicated by early data in Figure 2d, viruses were largely cleared by 4 dpi in mice treated with B5-D3 prophylaxis. Therefore, in this test, we did not examine virus titers in the recovered animals by the end of observation at 14 dpi. Instead, we examined plasma levels of virus-neutralizing antibodies in the survivors at the endpoint, which indeed supported that the 6-hours and 24-hours IN B5-D3 prophylaxis provided effective protection against the SARS-CoV-2 infection and resulted in minimal levels of neutralizing antibodies in plasma, as shown in Figure 2i.

Collectively, the body weight, survival, and antibody data all supported that 6-hour IN B5-D3 prophylaxis achieved the best efficacy. Hence, we performed comprehensive viral titer and profiling analysis at early time points like 1 dpi, 2 dpi, and 4 dpi, focusing only on the 6-hour IN B5-D3 prophylaxis. This works also included B5-D3-LALA control to examine viral titers, host immune responses, and underlying mechanisms (Figure 3,4).

We agree with the reviewer that it would be more comprehensive if our experiments could include indepth analysis of the 24-hours IN B5-D3 prophylaxis group. However, due to limited capacity of animal service, we chose to focus on the best-performing group as a representative treatment to study the underlying mechanisms.

(6) In Figures 3b and 3c, viral spike (S) and nucleocapsid (N) RNA relative expression levels are quantified by qPCR. The results show significant individual variation within the B5-D3-LALA treatment group: one mouse exhibits high S and N expression, while the other two show low expression. Viral load levels are also inconsistent: two mice have high viral loads, and one has a low viral load. Due to this variability, the available data are insufficient to robustly support the conclusion.

We understand the reviewer’s concern on the variability within the B5-D3-LALA group. However, we have some reservations about the importance of further increasing the sample sizes in this test.

First, since viral gene transcription and viral particle levels represented different phases in viral life, they may follow different kinetics during infection progression and lead to variability. Second, we used different parts of the lung tissues from each mouse for extracting RNA and tissue homogenates, which were then used for detection of S/N expression and viral load levels, respectively. The uneven viral infection in the lung might also contribute to the variability. Furthermore, in this test, both our qPCR and viral load analysis data consistently demonstrated that the B5-D3-LALA was less effective than B5-D3, indicating that Fc function played an important role in supporting full protection by B5-D3 against lethal SAS-CoV-2 infections. This observation is also supported by other studies [12].

We appreciate the valuable feedback from the reviewer. In the revised manuscript, we have further clarified these observations on page 8, lines 192–194, and included alveolar thickening data on page 9, lines 202–204.

(7) Figure 3e: "H&E staining indicated alveolar thickening in all groups," including the Mock group. Since the Mock group did not receive virus or active drug treatment, this observed change may result from local tissue reaction induced by the intranasal inoculation procedure itself, rather than specific immune activation. A control group (no manipulation) should be set to rule out potential confounding effects of the experimental procedure on tissue morphology, thereby allowing a more accurate assessment of the drug's effects.

We thank the reviewer for his insightful comments and suggestions.

We have further examined our H&E staining and quantified alveolar thickening in different treatment groups. Indeed, the data suggested a transient alveolar thickening in the mock group at 1 dpi, which was improved at 2 dpi. This observation supports that the intranasal procedure itself indeed caused a transient alveolar thickening, that was evident at 1 dpi but disappeared at 2 dpi.

Notably, moderate alveolar thickening was found to be persistent in the B5-D3-treated mice till the end point at 4 dpi. Whereas the PBS groups with intensive SARS-CoV-2 infection progressively developed severe structural damage and showed much stronger alveolar thickening than B5-D3 or mock groups at 4 dpi. Consistent with the partial protection by B5-D3-LALA, histological analysis of lung samples in this group revealed severer yet heterogenous alveolar thickening. These observations suggested that -6h IN B5-D3 treatment prevented tissue damage brought by infection with minimal yet efficient immune activation.

In the revised manuscript, we have included the quantitation results of alveolar thickening on page 9, lines 200–204 and presented the data in new Supplementary Fig. 7.

(8) In Supplementary Figure 11b, a considerable number of alveolar macrophages (AMs) are observed in both the PBS and B5-D3 groups. This makes it difficult to determine whether the observed accumulation is specifically induced by B5-D3.

We thank the reviewer for pointing out this issue.

In this experiment, the cell populations examined in previous Supplementary Fig. 11b and Fig. 5h are different, though graphs appear similar.

Supplementary Fig. 11b (new Supplementary Fig. 12b) showed the analysis among CD45+ immune cells, regardless of B5-D3-AF750 signal. The dominance of AMs among immune cell populations is a normal physiological feature of BALF cells. To make this clear, we have added new data of BALF cells from untreated mice in the revised manuscript and new Supplementary Fig. 12b.

Fig. 5h displayed for cell type analysis among the CD45+ B5-D3-AF750+ cells —only CD45+ immune cells that took up the AF750-labeled B5-D3.

To enhance clarity, in the revised manuscript, we have amended the labels as CD45+ B5-D3-AF750+ in Figure 5h (and similarly in revised Supplementary Fig. 13), to differentiate the data from that in CD45+ cells shown in the revised Supplementary Fig. 12b.

(9) In the flow cytometry experiment shown in Figure 5, the PBS control group is not labeled with AF750, which necessarily results in a value of zero for "B5-D3+ cells" on the y-axis. An appropriate control (e.g., hIgG1-Fc labeled with AF750) should be included.

We thank the reviewer for his valuable question.

In this experiment, we intended to analyze all immune cells with positive AF750 signals, to identify the major immune cell types that took up AF750-B5-D3 as the candidate cells responsible for the observed activation of innate immunity. Hence, here we deliberately set PBS vehicle treatment without AF750 signal as the control group for gating.

This analysis aimed to provide an overall picture of immune cell types that actively take up ACE2 decoy, likely via Fc receptor-mediated binding. Control IgG1 labeled with AF750, with an Fc region, may show similar profile and biodistribution among BALF immune cells, which, therefore, was not examined as control for gating.

Instead, in the revised manuscript, we have added new analysis results comparing the efficiencies of B5-D3 and IgG1 in mediating pseudovirus uptake in THP-1-derived macrophages. IgG1 isotype control was examined to address ACE2-specific effect. Indeed, we observed no pseudovirus uptake based on p24 signal, in the IgG1 treated samples, indicating that the presence of B5-D3 is crucial for efficient pseudovirus uptake in macrophages due to the sACE2-spike affinity. These results have been added on page 13 lines 310–316 in the main text, and the relevant data was presented in new Supplementary Fig. 17.

(10) The Methods section: a more detailed description of the experimental procedures involving HIV p24 and SARS-CoV-2 should be included.

We thank the reviewer for pointing out this weakness.

In the revised manuscript, we have provided further details of the relevant experimental procedures in the Materials and Methods part, on page 21, lines 507–517.

**Reviewer #3 (Public review):**
Strengths:The core strength of this study lies in its innovative demonstration that an engineered sACE2-Fc fusion redirects virus-decoy complexes to Fc-mediated phagocytosis and lysosomal clearance in macrophages, revealing a distinct antiviral mechanism beyond traditional neutralization. Its complete prophylactic protection in animal models and precise targeting of airway phagocytes establish a novel therapeutic paradigm against SARS-CoV-2 variants and future respiratory viruses.

We thank the reviewer for his recognition and positive comments on our study.

Weaknesses:The study attributes complete antiviral protection to Fc-mediated phagocytic clearance, a central claim that requires more rigorous experimental validation. The observation that abrogating Fc functions compromises protection could be confounded by potential alterations in the protein's stability, half-life, or overall structure. To firmly establish this mechanism, it is crucial to include a control molecule with a mutated Fc region that lacks FcγR binding while preserving the Fc structure itself. Without this critical control, the conclusion that phagocytic clearance is the primary mechanism remains inadequately supported.

We thank the reviewer for his insightful comments and suggestions.

The L234A/L235A mutations in human IgG1 Fc region are most widely used to abolish its FcγR binding and Fc effector functions [13]. In this study, we have used B5-D3-LALA in the in vivo infection experiments in K18-hACE2 mice, as the control molecule that lacks FcγR binding while preserving the Fc structure (Figure 3, 4).

To address the reviewer’s concern, we further performed new analysis comparing the efficiencies of different versions of B5-D3 in mediating pseudovirus uptake in THP-1-derived macrophages. In this test, B5-D3-LALA and B5-D3 were examined side-by-side to address the role of Fc effector functions in the phagocytosis process. Meanwhile, IgG1 isotype control was examined to address ACE2-specific effect. Indeed, we detected significant reduction of pseudovirus uptake based on p24 signal, in the B5D3-LALA treated samples compared to those receiving B5-D3. This decreased pseudoviral uptake correlated with the loss of Fc-mediated effector functions in B5-D3-LALA, indicating the involvement of Fc functions in efficient macrophage uptake of B5-D3-virus complex.

In the revised manuscript, we have included these results on page 13 lines 310–316 in the main text and presented relevant data in Supplementary Fig. 17.

The strategy of deliberately targeting virus-decoy complexes to phagocytes via Fc receptors inherently raises the question of Antibody-Dependent Enhancement (ADE) of disease. While the authors demonstrate a lack of productive infection in macrophages, this only addresses one facet of ADE. The risk of Fc-mediated exacerbation of inflammation (ADE) remains a critical concern. The manuscript would be significantly strengthened by a direct discussion of this risk and by including data, such as cytokine profiling from treated macrophages, to more comprehensively address the safety profile of this approach.

(1) We thank the reviewer for his insightful comments and suggestions regarding the ADE issue.

Indeed, Antibody-Dependent Enhancement (ADE) of viral infection is a critical concern when developing the ACE2 decoy strategy. In this study, we have carefully examined the relevant risk based on our data from various in vitro and in vivo assays.

In our in vivo infection experiments, all B5-D3 prophylaxis and treatment groups, regardless of the administration times and routes, showed improved outcomes like less body-weight loss and better survival, compared to the PBS control groups (Figure 2). None of these treatment groups demonstrated worsened infections, indicating that ADE phenomenon was not occurring or did not play a major role during the B5-D3 treatments. Instead, moderate immune activation was observed in the lung of B5-D3 treated mice, which occurred much earlier but was milder compared to that in the PBS groups, and may reflect responses that lead to the efficient early clearance of viruses without observable symptoms (Figure 3 and 4).

In our in vitro assays shown in Figure 6, B5-D3 treatments in epithelial or non-immune cell models (hACE2-Galu-3 and hACE2-293T) significantly blocked the entry of pseudovirus into cells and yielded much reduced luciferase signals (Figure 6d–g). Whereas in the THP-1-derived macrophages, although the presence of B5-D3 largely enhanced the entry of SARS-CoV-2 pseudovirus into cells (Figure 6a,b), it did not result in active infection and produced no luciferase signal (Figure 6g). These results were robustly reproducible, indicating that pseudoviruses did not successfully release its genome RNA and viral proteins (like RTase and integrases) after entering macrophages. Instead, colocalization analysis of p24 (pseudoviruses), sACE2-Fc (B5-D3), and LAMP1 (lysosome) signals suggested probability of pseudovirus degradation in endosomes/lysosomes after cell entry (Figure 6a,c). Consistently, examination of the macrophages that had taken up pseudovirus showed that the Spike (S) proteins from the pseudovirus particles were not cleaved to release S2’ fragment at a distinct smaller size (Figure 6h). As the cleavage of S protein in host cells is critical for effective membrane fusion, it is essential and regarded as hallmark for successful viral entry and escape from endosome. Collectively, these data consistently indicated that the SARS-CoV-2 pseudoviruses were degraded directly in lysosomes after entering macrophages, showing no sign of ADE.

(2) We thank the reviewer for his valuable suggestion and have performed RNA-seq analysis to profile immune responses in the treated macrophages.

We performed RNA-Seq analysis to investigate major transcriptional changes in THP-1-derived macrophages after the pseudovirus infection, with or without B5-D3 treatments. Although no individual genes fulfilled the cutoff threshold of significant up-/down-regulation, we observed antiviral responses in the pseodovirus-B5-D3 treated samples by GSEA (new Supplementary Fig. 18). This observation indicated that the B5-D3 treatment and subsequent cell-entry of pseudovirusB5-D3 complexes into macrophages induced immune activation at moderate levels, but not evoking strong immune responses that can be harmful to the host.

In the revised manuscript, we have included the new RNA-seq analysis results on macrophage infection tests on page 13 lines 317–322 and page 14 lines 323–325 in the main text and presented the relevant data in the new Supplementary Fig. 18. Furthermore, we agree that ADE is a critical issue and have further enriched our discussion on page 17 lines 415–417, to emphasize that the risk for ADE should be thoroughly evaluated to further develop the decoy strategy for human use.

The exclusive use of the K18-hACE2 mouse model, which exhibits severe disease, limits the generalizability of the findings. The "complete protection" observed may not translate to models with more robust and naturalistic immune responses or to human physiology.

We thank the reviewer for pointing out the limitation of the mouse model used.

(1) Given that wild type mice are not susceptible to SARS and SARS-CoV-2 infection, transgenic mice have been generated to express hACE2, through various designs and strategies, serving as models for viral infection and drug development. However, many of these hACE2 transgenic mouse models exhibit mild infections due to moderate hACE2 levels, failing to develop the severity observed in SARS and COVID patients [14].

(2) The K18-hACE2 transgenic mouse line (B6. Cg-Tg(K18-ACE2)2Prlmn/J, Jackson Laboratory) used in our study carries multiple copies of K18-hACE2 transgene cassette [15]. Compared to other hACE2 transgenic mouse models, this K18-hACE2 line shows higher expression of hACE2 in airway and other epithelia and supports severer infections by both SARS and SARS-CoV2 viruses, successfully causing lethality [16]. Hence, K18-hACE2 mice is a widely used model to study SARS and SARS-CoV2 virus infections and drug developments.

(3) We agree that K18-hACE2 mice is a relatively weak transgenic line with poor productivity. However, it demonstrates best susceptibility to SARS-CoV-2 infection among established mouse models. In this study, we observed robust responses to SARS-CoV-2 infection in both aged and young cohorts, with all infected mice consistently demonstrating significant body weight loss during 4 dpi to 7 dpi (the PBS groups in Figure 2b, g)

We agree with the reviewer that it would be more convincing to assess the efficacy of B5-D3 using additional animal models. However, we have some reservations about the importance of these additional tests. First, the generality of ACE2-Fc decoy concept and its efficacy have been reported in other studies using various models [17,18]. Moreover, different transgenic mice or animal models exhibit distinct kinetics in the pathogenesis process and immune responses to SAS-CoV-2 infections, which differ from that in human patients at varied aspects. Hence, given the limited capacity of animal facility, we chose to focus on the K18-hACE2 mice that have demonstrated most robust and convincing infection data, to investigate the potential of B5-D3 administered through various strategies as well as the underlying mechanisms for the full protection observed in IN prophylaxis.

In the revised manuscript, we have further enriched our discussion regarding this limitation, on page 17 lines 417–422.

Furthermore, the lack of data on circulating SARS-CoV-2 variants is a concern

We thank the reviewer for his valuable comment.

In this study, we have demonstrated the viral neutralization capacity of B5-D3, as a representative of the enhanced sACE2 decoy, using multiple pseudoviruses and authentic SARS-CoV-2, which collectively covered eleven variants (up to Omicron strains). Our results from both in vitro neutralization and PRNT experiments confirmed the robust resilience of B5-D3 against viral evolution (Figure 1c–g). This observation aligns well with other studies and is broadly supported by various investigations, as was pointed out below by the reviewer.

Furthermore, studies on viral evolution have observed a robust trend that later-emerging SARS-CoV-2 variants exhibit a higher affinity for the ACE2 receptor, enhancing their infectivity and transmissibility [19]. Therefore, it is unlikely for a newly emerged SARS-CoV-2 variant to escape from B5-D3mediated neutralization.

Collectively, all evidence consistently supports the principle of decoy design, B5-D3 (or other effective ACE2 decoys) possess the intrinsic ability to neutralize new circulating SARS-CoV-2 variants, as long as the virus variants rely on ACE2 receptor for cell entry. Hence, although further tests on circulating viral variants would add strengths to our study, the significance of this additional data may be limited.

In the revised manuscript, we have further addressed this concern in the discussion, on page 16 lines 394–397.

The concept of sACE2-Fc fusion proteins as decoy receptors is not novel, and numerous similar constructs have been previously reported. The manuscript would benefit from a clearer demonstration of how the optimized B5-D3 mutant represents a significant advance over existing sACE2-Fc designs.

We thank the reviewer for his valuable comments.

Indeed, previous research has reported multiple ACE2 mutations to enhance its binding to spike proteins and neutralization against SARS-CoV-2. However, combining ACE2 mutations based on *in silico* predictions to both enhance spike binding and eliminate the ACE2 enzymatic activity resulted in accumulated burdens. For instance, ACE2 decoy candidates with up to five mutations like K31F/N33D/H34S/E35Q/H345L [8] and L79F/M82Y/Q325Y/H374A/H378A [12] have demonstrated excellent potency to neutralize SARS-CoV-2 in both in vitro and in vivo assays. However, the extensive mutations could be associated with structural instability and reduced production efficiency [8,12]. Furthermore, the high mutation loads increase risks for immunogenicity, which is a critical issue in future clinical applications. Corroboratively, Urano et al. detected in vitro T cell stimulation elicited by the L79F mutation, whereas the T92Q mutation (included in our decoy design) showed much lower immunogenicity and enhanced spike binding affinity [20].

In our ACE2 decoy design, we incorporated only two mutations (like T92Q and H374N in B5-D3) to enhance neutralization potency while eliminating enzymatic activity, resulting in simplest ACE2 mutants desired for engineering enhanced decoy. B5-D3, as one representative, not only exhibited minimal mutation-related risks (Supplementary Fig. 2i) but also top-level neutralization potencies among all candidate mutants tested (Figure 1, Supplementary Fig. 2f,g and Supplementary Fig. 3). To further address the safety of B5-D3 for in vivo use, we have performed prolonged in vivo overexpression of B5-D3 ACE2 decoy through AAV delivery in immune-competent K18-hACE2 mice, which indeed showed no sign of RAS disturbance or immune infiltration causing tissue damage. (In the revise manuscript, we have included these new results on page 5 lines 118–122 and page 6 lines 123–135 in the main text and presented the data in new Supplementary Fig. 4).

Therefore, instead of demonstrating advantage over existing sACE2-Fc designs, our study used the optimized B5-D3 as a representative ACE2 decoy of top performers, to systematically examined various administration strategies as well as the underlying mechanisms for the full protection observed in IN prophylaxis. Aligned with this effort, our study identified 6-hours IN prophylaxis as the most effective regimen to confer complete protection against SARS-CoV-2 infection in K18-hACE2 mice. Further investigation through transcriptomics, bio-distribution, and phagocytosis analysis revealed that IN-delivered B5-D3 not only neutralizes viruses but also engaged airway phagocytes to promote early viral clearance and host immune activation, uncovering a distinct antiviral mechanism for the universal “decoy strategy” to combat unknown air-borne respiratory virus in the future.

In the revised manuscript, we have further clarified our focus on using B5-D3 as a “representative” of ACE2 decoy on page 4 line 84, page 5 line 109, page 14 line 333, and page 15 line 360.

A direct comparative analysis with previously published benchmarks, particularly in terms of neutralizing potency, Fc effector function strength, and in vivo efficacy, is necessary to establish the incremental value and novelty of this specific agent.

We thank the reviewer for his valuable comments.

Indeed, our study has aimed to address this concern and made partial progress through in vitro neutralization assays (Figure 1b and Supplementary Fig. 2c,d,f,g). Our results from the limited yet meaningful comparisons with the sACE2 lacking Fc domain and selected sACE2-Fc mutants published/proposed previously clearly demonstrated “substantial enhancement through Fc-fusion” (Supplementary Fig. 1d) and modest improvement from protein mutagenesis at ACE2-Spike interaction interface” (Figure 1b and Supplementary Fig. 2c,d,f,g).

Based on the results from our various neutralization assays, we chose B5-D3 as a representative of enhanced decoy for in vivo infection, which identified 6-hours IN prophylaxis to confer complete protection against infection, demonstrating significant impact of administration strategies on in vivo efficacy of B5-D3 (Figure 2). Subsequent analysis further uncovered intriguing phenomena regarding the cellular distribution of IN-administered B5-D3 and the early immune activation triggered in the lung, which underlies the full protection by IN prophylaxis and represents an important novelty of this study.

We agree with the reviewer that further analysis with additional benchmark versions would enhance the value of this study, but we have reservation regarding the importance. To enhance clarity, in the revised manuscript, we have further emphasized our study focus on using B5-D3 as a representative ACE2 decoy throughout the text and enriched the discussion on page 15 line 348–365.

References

(1) Ku Z, Xie X, Hinton PR, Liu X, Ye X, Muruato AE, Ng DC, Biswas S, Zou J, Liu Y, Pandya D, Menachery VD, Rahman S, Cao Y-A, Deng H, Xiong W, Carlin KB, Liu J, Su H, Haanes EJ, Keyt BA, Zhang N, Carroll SF, Shi P-Y & An Z. Nasal delivery of an IgM offers broad protection from SARS-CoV-2 variants. Nature 595, 718-723 (2021).

(2) Liu J, Mao F, Chen J, Lu S, Qi Y, Sun Y, Fang L, Yeung ML, Liu C, Yu G, Li G, Liu X, Yao Y, Huang P, Hao D, Liu Z, Ding Y, Liu H, Yang F, Chen P, Sa R, Sheng Y, Tian X, Peng R, Li X, Luo J, Cheng Y, Zheng Y, Lin Y, Song R, Jin R, Huang B, Choe H, Farzan M, Yuen KY, Tan W, Peng X, Sui J & Li W. An IgM-like inhalable ACE2 fusion protein broadly neutralizes SARSCoV-2 variants. Nat Commun 14, 5191 (2023).

(3) Guo H, Cho B, Hinton PR, He S, Yu Y, Ramesh AK, Sivaccumar JP, Ku Z, Campo K, Holland S, Sachdeva S, Mensch C, Dawod M, Whitaker A, Eisenhauer P, Falcone A, Honce R, Botten JW, Carroll SF, Keyt BA, Womack AW, Strohl WR, Xu K, Zhang N, An Z, Ha S, Shiver JW & Fu T-M. An ACE2 decamer viral trap as a durable intervention solution for current and future SARS-CoV. Emerging Microbes & Infections 12, 2275598 (2023).

(4) Keyt BA, Baliga R, Sinclair AM, Carroll SF & Peterson MS. Structure, Function, and Therapeutic Use of IgM Antibodies. Antibodies 9, 53 (2020).

(5) Chan KK, Dorosky D, Sharma P, Abbasi SA, Dye JM, Kranz DM, Herbert AS & Procko E. Engineering human ACE2 to optimize binding to the spike protein of SARS coronavirus 2. Science 369, 1261-1265 (2020).

(6) Guy JL, Jackson RM, Jensen HA, Hooper NM & Turner AJ. Identification of critical active-site residues in angiotensin-converting enzyme-2 (ACE2) by site-directed mutagenesis. The FEBS Journal 272, 3512-3520 (2005).

(7) Payandeh Z, Rahbar MR, Jahangiri A, Hashemi ZS, Zakeri A, Jafarisani M, Rasaee MJ & Khalili S. Design of an engineered ACE2 as a novel therapeutics against COVID-19. Journal of Theoretical Biology 505, 110425 (2020).

(8) Glasgow A, Glasgow J, Limonta D, Solomon P, Lui I, Zhang Y, Nix MA, Rettko NJ, Zha S, Yamin R, Kao K, Rosenberg OS, Ravetch JV, Wiita AP, Leung KK, Lim SA, Zhou XX, Hobman TC, Kortemme T & Wells JA. Engineered ACE2 receptor traps potently neutralize SARS-CoV2. Proceedings of the National Academy of Sciences 117, 28046-28055 (2020).

(9) Lei C, Qian K, Li T, Zhang S, Fu W, Ding M & Hu S. Neutralization of SARS-CoV-2 spike pseudotyped virus by recombinant ACE2-Ig. Nature Communications 11, 2070 (2020).

(10) Maciuba S, Bowden GD, Stratton HJ, Wisniewski K, Schteingart CD, Almagro JC, Valadon P, Lowitz J, Glaser SM, Lee G, Dolatyari M, Navratilova E, Porreca F & Riviere PJM. Discovery and characterization of prolactin neutralizing monoclonal antibodies for the treatment of female-prevalent pain disorders. MAbs 15, 2254676 (2023).

(11) Dwivedi V, Shivanna V, Gautam S, Delgado J, Hicks A, Argonza M, Meredith R, Turner J, Martinez-Sobrido L, Torrelles JB & Kulkarni V. Age associated susceptibility to SARS-CoV-2 infection in the K18-hACE2 transgenic mouse model. Geroscience 46, 2901-2913 (2024).

(12) Chen Y, Sun L, Ullah I, Beaudoin-Bussières G, Anand SP, Hederman AP, Tolbert WD, Sherburn R, Nguyen DN, Marchitto L, Ding S, Wu D, Luo Y, Gottumukkala S, Moran S, Kumar P, Piszczek G, Mothes W, Ackerman ME, Finzi A, Uchil PD, Gonzalez FJ & Pazgier M. Engineered ACE2-Fc counters murine lethal SARS-CoV-2 infection through direct neutralization and Fc-effector activities. Science Advances 8, eabn4188 (2022).

(13) Lund J, Winter G, Jones PT, Pound JD, Tanaka T, Walker MR, Artymiuk PJ, Arata Y, Burton DR, Jefferis R & Woof JM. Human Fc gamma RI and Fc gamma RII interact with distinct but overlapping sites on human IgG. The Journal of Immunology 147, 2657-2662 (1991).

(14) Lutz C, Maher L, Lee C & Kang W. COVID-19 preclinical models: human angiotensinconverting enzyme 2 transgenic mice. Hum Genomics 14, 20 (2020).

(15) McCray PB, Pewe L, Wohlford-Lenane C, Hickey M, Manzel L, Shi L, Netland J, Jia HP, Halabi C, Sigmund CD, Meyerholz DK, Kirby P, Look DC & Perlman S. Lethal Infection of K18hACE2 Mice Infected with Severe Acute Respiratory Syndrome Coronavirus. Journal of Virology 81, 813-821 (2007).

(16) Oladunni FS, Park JG, Pino PA, Gonzalez O, Akhter A, Allue-Guardia A, Olmo-Fontanez A, Gautam S, Garcia-Vilanova A, Ye C, Chiem K, Headley C, Dwivedi V, Parodi LM, Alfson KJ, Staples HM, Schami A, Garcia JI, Whigham A, Platt RN, 2nd, Gazi M, Martinez J, Chuba C, Earley S, Rodriguez OH, Mdaki SD, Kavelish KN, Escalona R, Hallam CRA, Christie C, Patterson JL, Anderson TJC, Carrion R, Jr., Dick EJ, Jr., Hall-Ursone S, Schlesinger LS, Alvarez X, Kaushal D, Giavedoni LD, Turner J, Martinez-Sobrido L & Torrelles JB. Lethality of SARS-CoV-2 infection in K18 human angiotensin-converting enzyme 2 transgenic mice. Nat Commun 11, 6122 (2020).

(17) Urano E, Itoh Y, Suzuki T, Sasaki T, Kishikawa JI, Akamatsu K, Higuchi Y, Sakai Y, Okamura T, Mitoma S, Sugihara F, Takada A, Kimura M, Nakao S, Hirose M, Sasaki T, Koketsu R, Tsuji S, Yanagida S, Shioda T, Hara E, Matoba S, Matsuura Y, Kanda Y, Arase H, Okada M, Takagi J, Kato T, Hoshino A, Yasutomi Y, Saito A & Okamoto T. An inhaled ACE2 decoy confers protection against SARS-CoV-2 infection in preclinical models. Sci Transl Med 15, eadi2623 (2023).

(18) Higuchi Y, Suzuki T, Arimori T, Ikemura N, Mihara E, Kirita Y, Ohgitani E, Mazda O, Motooka D, Nakamura S, Sakai Y, Itoh Y, Sugihara F, Matsuura Y, Matoba S, Okamoto T, Takagi J & Hoshino A. Engineered ACE2 receptor therapy overcomes mutational escape of SARS-CoV-2. Nature Communications 12, 3802 (2021).

(19) Cho MJ, Been NR & Son H. From Alpha to Omicron: Structural Insights into SARS-CoV-2 RBD Evolution and ACE2 Binding. European Journal of Public Health 35(2025).

(20) Urano E, Itoh Y, Suzuki T, Sasaki T, Kishikawa J-i, Akamatsu K, Higuchi Y, Sakai Y, Okamura T, Mitoma S, Sugihara F, Takada A, Kimura M, Nakao S, Hirose M, Sasaki T, Koketsu R, Tsuji S, Yanagida S, Shioda T, Hara E, Matoba S, Matsuura Y, Kanda Y, Arase H, Okada M, Takagi J, Kato T, Hoshino A, Yasutomi Y, Saito A & Okamoto T. An inhaled ACE2 decoy confers protection against SARS-CoV-2 infection in preclinical models. Science Translational Medicine 15, eadi2623 (2023).